# The parenting hub of the hypothalamus is a focus of imprinted gene action

Matthew J. Higgs[1], Anna E. Webberley[1], Alasdair J. Allan[2], Moaz Talat[2], Rosalind M. John[3], Anthony R. Isles[1]*

1 Behavioural Genetics Group, Centre for Neuropsychiatric, Genetics and Genomics, Neuroscience and Mental Health Innovation Institute, Cardiff University, Cardiff, United Kingdom, 2 The Mary Lyon Centre, MRC Harwell, Didcot, United Kingdom, 3 School of Biosciences, Cardiff University, Cardiff, United Kingdom

* islesar1@cardiff.ac.uk

**Data Availability Statement:** The sequencing data used in this project were acquired through previously published, publicly available resources, the details of which can be found in the Supplemental Materials. Briefly, the datasets used

## Abstract

Imprinted genes are subject to germline epigenetic modification resulting in parental-specific allelic silencing. Although genomic imprinting is thought to be important for maternal behaviour, this idea is based on serendipitous findings from a small number of imprinted genes. Here, we undertook an unbiased systems biology approach, taking advantage of the recent delineation of specific neuronal populations responsible for controlling parental care, to test whether imprinted genes significantly converge to regulate parenting behaviour. Using single-cell RNA sequencing datasets, we identified a specific enrichment of imprinted gene expression in a recognised "parenting hub", the galanin-expressing neurons of the preoptic area. We tested the validity of linking enriched expression in these neurons to function by focusing on MAGE family member L2 *(Magel2)*, an imprinted gene not previously linked to parenting behaviour. We confirmed expression of *Magel2* in the preoptic area galanin expressing neurons. We then examined the parenting behaviour of *Magel2*-null$^{(+/p)}$ mice. *Magel2*-null mothers, fathers and virgin females demonstrated deficits in pup retrieval, nest building and pup-directed motivation, identifying a central role for this gene in parenting. Finally, we show that *Magel2*-null mothers and fathers have a significant reduction in POA galanin expressing cells, which in turn contributes to a reduced *c-Fos* response in the POA upon exposure to pups. Our findings identify a novel imprinted gene that impacts parenting behaviour and, moreover, demonstrates the utility of using single-cell RNA sequencing data to predict gene function from expression and in doing so here, have identified a purposeful role for genomic imprinting in mediating parental behaviour.

## Author summary

Genomic imprinting is a fascinating phenomenon that affects a small sub-group of the approximately 22,000 found in mammals. Unlike most genes that are equally expressed from both inherited parental copies (or alleles), so called imprinted genes are only expressed from one inherited allele, and this is usually fixed so that some imprinted genes are only active from the maternal copy, whereas others are only active from the paternal copy. This silencing of one of the parental copies makes genomic imprinting and

are GSE113576 (Gene Expression Omnibus); SRP135960 (Sequence Read Archive); GSE87544 (Gene Expression Omnibus). All data generated in this experiment is provided as Supplemental Data and in the following Open Science Framework repository (https://osf.io/jx7kr/). Custom R scripts are provided as Supplemental Code and are available at https://github.com/MJHiggs/IG-Single-Cell-Enrichment.

**Funding:** This work was supported by a Wellcome Trust PhD studentship (220090/Z/20/Z) to ARI, MJH and RMJ. Furthermore, we are extremely grateful to the Foundation for Prader-Willi Research with the Mary Lyon Centre, International Facility for Mouse Genetics, at MRC Harwell for generating the mouse model and producing the experimental cohorts used in this study (MC_UP_2201/1). The funders had no role in study design, data collection and analysis, decision to publish, or preparation of the manuscript.

**Competing interests:** The authors have declared that no competing interests exist.

evolutionary conundrum and the best way to understand why imprinted genes exist is to investigate the physiologies upon which they impact. Here we investigated imprinted gene expression in the brain circuitry that controls parental behaviours in mammals. We show that as a group the imprinted genes are disproportionately represented in the gene expression profile of the key neurons in this circuitry. We then tested this approach by showing that loss expression of a gene called *Magel2* that was one of those imprinted genes identified in this brain circuitry, leads to deficits in parental behaviour in mice. Taken together with previous work, our findings indicate that genomic imprinting plays a particularly important role in the control of parenting behaviour.

## Introduction

Imprinted genes (IGs) demonstrate a preferential or exclusively monoallelic expression from either the maternal or paternal allele in an epigenetically predetermined manner (a parent-of-origin effect, POE). To date approximately 260 imprinted genes, demonstrating biased allelic expression and/or associated with a parental-specific epigenetic mark, have been identified in the mouse (~230 in humans) [1, 2]. This epigenetic regulation makes genomic imprinting an evolutionary puzzle, as many of these genes are effectively haploid and thereby negate many of the benefits of diploidy [3]. Consequently, there is a great deal of interest in the functional role of imprinted genes and the physiologies upon which they impact.

In adult mice, the brain is consistently shown to be one organ where a large number of genes are imprinted [4–6] and studies of mice carrying manipulations of individual imprinted genes have suggested a wide range of behavioural roles [2, 7]. These, and other studies [8, 9], have suggested a particular focus on the hypothalamus for genomic imprinting, and a number of key hypothalamic-related behaviours, such as feeding [10] and sleep [11], have been repeatedly linked to imprinted genes. Another well-known associated behaviour is maternal caregiving and, to date, four paternally expressed imprinted genes have been shown to impact parenting when disrupted: *Mest/Peg1* [12], *Peg3* [13], *Dio3* [14] and *Peg13* [15]. In all four cases, mutant mothers raising functionally WT litters had impaired maternal behaviour. These independent findings have led to the suggestion that maternal care is a physiological focus for imprinted genes [16–18] and potentially relevant to the evolution of genomic imprinting [19–21]. However, whether the effect on maternal care of these four genes represents serendipitous, coincidental findings, or is indicative of a convergent role for imprinted genes has not been formally tested.

The neural circuitry underlying maternal behaviour has now been substantially determined in mice. The work of Numan and colleagues [22, 23] identified the core neural circuitry necessary for parenting and found a hub region, the medial preoptic area (MPOA) in the hypothalamus, which was essential for parenting behaviour [24, 25]. When activated optogenetically, the MPOA could produce parenting behaviour on demand [26] even in animals not normally capable [27]. Recent work has identified the specific neuron-types within this circuitry, showing a critical role for the galanin expressing neurons within the preoptic area as the hub neurons, receiving and sending input to many other brain regions in order to produce the specific facets of parenting behaviour in mothers, fathers and virgins [27, 28]. Significantly, modern extensive single cell RNA sequencing and in-situ work [29] has resolved the neural populations of the POA and has refined the population of neurons with the largest *c-Fos* response to parenting behaviour in mothers, fathers and virgin females–*Gal*-expressing neurons co-expressing *Th* and *Calcr*, and *Brs3*.

Utilising a variety of publicly available single cell transcriptomic data, we have previously demonstrated that imprinted genes show over-representation in the adult mouse brain, and more specifically gene set enrichment in the neurons and neuroendocrine cells of the hypothalamus [30]. We also found that at multiple levels of analysis (i.e., between neurons in the whole brain and between neurons from the hypothalamus) similar neural subpopulations were enriched for imprinted genes, specifically GABAergic neurons expressing either *Agrp/Npy*, *Avp/Nms*, *Ghrh*, or *Gal*. The enrichment in galanin expressing GABAergic neurons of the hypothalamus were of particular interest to us as this population of neurons could potentially contain the parenting associated *Gal/Th/Calcr/Brs3* neurons.

Here, we aimed to systematically investigate the role imprinted genes play in parenting behaviour. Using single-cell RNAseq data, we show imprinted gene expression to be enriched in the specific parenting-associated *Gal*-expressing neurons of the POA at multiple resolutions. Then, to test the validity of inferring function from expression in this manner, we examined parenting behaviour in mice null for one of the imprinted genes (*Magel2*) identified from our analyses, but which had not previously been linked with provision of parental care. First, we confirmed the elevated expression of *Magel2* in these POA-*Gal* neurons using RNAscope, then we assessed the parenting behaviour of *Magel2*-null mice using the retrieval-nest building and three chambers assessments. Finally, we used RNAscope to assess the impact of knocking out *Magel2* on POA galanin levels and upon the POA *c-Fos* response when exposing mice to pups. Together, our data conclusively show that parental care is indeed a physiological focus for genomic imprinting and suggest a new mechanism by which these genes could be affecting this behaviour.

## Results

### Imprinted gene expression is enriched in the parenting associated Gal-neurons of the MPOA

To assess the role of IGs in the galanin enriched neurons of the POA specifically active during parenting, we analysed the highest resolution mouse POA dataset available ([29]. Enrichment analysis (Table A in S1 Text) found imprinted gene expression to be over-represented in two of the 66 POA neuron subpopulations identified [29].: i35: *Crh/Tac2* (9/74 IGs, $q = 0.0126$) and i16:*Gal/Th* (15/74 IGs, $q = 0.0026$). Using MERFISH, the single cell population—i16:*Gal/Th*—was found [29] to be a composite of two neural populations (i16:*Gal/Th* and i14:*Gal/Calcr*), both of which significantly expressed *c-Fos* following parenting behaviour in mice. In our analysis the representative single cell population—i16:*Gal/Th*—was the top hit for enrichment of imprinted genes in the POA, and one of the only neuron subtypes in which imprinted genes displayed a higher mean fold change than the rest of the genes. Fig 1A displays the imprinted genes showing enrichment in this neuron type.

To further explore this finding, we identified those imprinted genes that were expressed highly in relevant GABAergic galanin neurons at multiple resolutions. The mouse brain atlas [31] resolved the entire murine nervous system into over 200 unique neuronal subpopulations. We previously demonstrated an imprinted gene enrichment in 11 hypothalamic neuron subpopulations. One of the top hits was TEINH3 which was the best match for the POA galanin neurons as it localized to the BNST/POA and expressed *Gal*, *Calcr* and *Brs3* amongst the top 20 marker genes (Table B in S1 Text and S1 Table). The imprinted genes making up that enrichment in TEINH3 are highlighted in Fig 1B. In a separate study [32], neurons isolated from the just the hypothalamus were resolved into 33 neuronal subpopulations. Of interest were two galanin enriched neuron types, GABA13 (*Gal*, *Slc18a2* and *Th* and GABA10 (*Gal*, *Calcr* and *Brs3)* (Table C in S1 Text). The imprinted genes highly expressed in both cell types

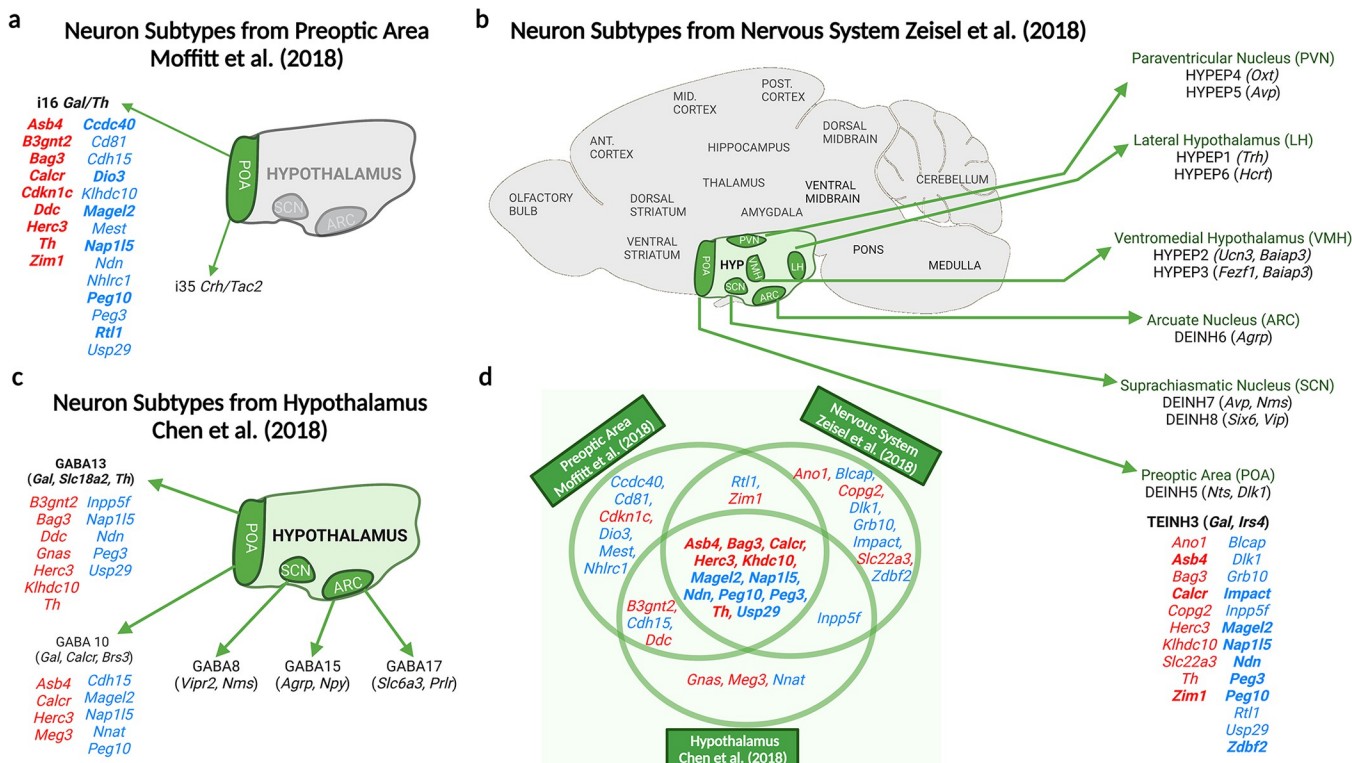

**Fig 1. Imprinted genes upregulated in galanin neuron subpopulations.** Imprinted genes are presented in red and blue which indicates their parent-of-origin expression; blue–PEGs, red–MEGs. Genes in black and present in parentheses are the neuronal markers for that cell type. A) Imprinted genes demonstrating > 150% expression level in the i16 *Gal/Th* neuronal subpopulation of the POA, shown to have elevated *c-Fos* during parenting behaviour, in the [29] dataset. Any genes showing ≥ 2-fold expression increase are boldened B) Hypothalamic neuron subpopulations found to have over-representation for imprinted genes in the [31] dataset. Imprinted genes with elevated expression in the *Gal/Calcr/Brs3* expressing population (TEINH3) are shown. Top 10 hits among imprinted genes are boldened. (C) Hypothalamic neuron subpopulations found to have over-representation for imprinted genes in the [32] dataset. Imprinted genes with elevated expression in the galanin-expressing population (GABA13 and GABA10 (not over-represented)) are shown. (D)Venn diagram highlighting imprinted genes present in more than one of the three investigations, 12 imprinted genes were over-represented in galanin enriched neurons in all three datasets. Created with Biorender.

are highlighted in Fig 1C. Marker genes for GABA10, GABA13 and TEINH13 are highlighted in S1 Table.

In summary, imprinted gene expression was found to be enriched in galanin expressing neurons of the hypothalamus when analysed at multiple resolutions. Furthermore, the parenting-associated galanin neurons make up part of this signal. For these parenting-specific *Gal/Th/Calcr/Brs3* neurons, 23 imprinted genes showed enriched expression (1/6 of the genes assessed). 12 imprinted genes were expressed in the relevant subpopulation at each level of analysis e.g. *Asb4*, *Calcr* (previously shown to be imprinted in the brain [33]), *Magel2*, *Ndn*, *Nap1l5*, *Peg3*, and *Peg10* while several more were expressed in 2/3 datasets e.g. *B3gnt2*, *Rtl1*, *Zim1*. This approach identified many imprinted gene not previously shown to have a role in parenting. We took one of these candidates, *Magel2*, forward for a parenting assessment.

## Magel2 expression pattern identify it as a candidate for behaviour characterisation

Prior to the parenting assessment, we sought to confirm the above findings by demonstrating an *in-situ* expression of *Magel2* in the parenting associated neurons of the MPOA [29]. We

proceeded by quantifying *Magel2* co-expression in both *Gal/Th* expressing neurons and *Gal/Calcr* expressing neurons in the POA.

Three-plex RNAscope was carried out on WT mouse brain sections taken at 100μm intervals through the preoptic area (9–10 sections per brain). 2 brains (1M/1F) were analysed using probes for *Gal*, *Th* & *Magel2* (Fig 2A) while 4 brains (2M/2F) were analysed using probes for *Gal*, *Calcr* & *Magel2* (Fig 2B) 2579 POA cells were identified as *Gal/Th* positive, representing 51% of the galanin positive cells while 3846 POA cells were identified as *Gal/Calcr* positive, representing 44% of galanin positive cells. There was clear co-expression of *Magel2* in both *Gal/Th* and *Gal/Calcr* cells (See Fig 2A–*Gal/Th* and 2B–*Gal/Calcr*). *Magel2* was found to be expressed in 88.3% of *Gal/Th* cells and 92.2% of *Gal/Calcr* cells, compared to the background POA *Magel2* expression rate of 57.5% of cells. We were underpowered to compare brains by sex, but values such as percentage of cells expressing *Magel2*, and average molecule counts were consistent from the two sexes suggesting no substantial differences between males and females.

Quantitative analysis of *Magel2* molecules in these galanin cell types (See Table D in S1 Text for all statistical summaries) found that significantly more *Magel2* molecules were present in *Gal/Th* cells (5.28 molecules) than all other cells in the POA (2.11 molecules, FC = 2.5, $P<0.001$, Mann-Whitney *U*-test). In order to further test the specificity of *Magel2* expression, we compared the POA *Gal/Th* cells to all other POA *Gal* positive cells, with the former having significantly more *Magel2* molecules (4.11 molecules, FC = 1.3). Similarly, there were more *Magel2* molecules in *Gal/Th* cells than in other POA *Th* positive cells (3.22 molecules, FC = 1.64) (See Fig 2C). We also restricted our analysis to only *Magel2* positive cells and found that there were still significantly more *Magel2* molecules in *Gal//Th* cells (5.99 molecules) compared to all other cells expressing *Magel2* (3.45 molecules, FC = 1.74, $P<0.001$, Mann-Whitney *U*-test). An identical finding was made when analysing *Gal/Calcr* cells. There were significantly more *Magel2* molecules in *Gal/Calcr* cells (5.81 molecules) than all other cells (1.59, FC = 3.66, $P<0.001$, Mann-Whitney *U*-test), all other *Gal* cells (3.06, FC = 1.89), all other *Calcr* cells (3.3 molecules, FC = 1.76) (See Fig 2D) and when only using *Magel2* positive cells in the analysis (6.3 vs. 2.91 molecules, FC = 2.16, $P<0.001$, Mann-Whitney *U*-test). Semi-quantitative H-scores were also calculated for all the comparisons listed above and *Gal/Th* and *Gal/Calcr* consistently displayed higher H-scores in all comparisons (See Table E in S1 Text). Fig A in S1 Text display histograms of these H-scores when compared to all of the rest of cells. Overall, these RNAscope studies validated our findings from the single-cell RNA-seq analysis that *Magel2* is expressed significantly higher in *Gal/Th/Calcr* cells compared to other cell types in the POA.

## Mice paternally inheriting inactivated Magel2 display parenting related deficits

We assessed parental behaviour in *Magel2*-null mice (paternal transmission of ablated allele), using three groups of mice capable of parenting behaviour: Primiparous Mothers, First-Time Fathers and Naïve Virgin Females. These groups were tested using a combined retrieval and nest building test paradigm [34] in which each animal had one hour to retrieve 3 scattered pups alongside reconstructing their deconstructed home nest. This was followed on a subsequent day by a Three-Chamber Pup-Preference test [35] in which the same 3 pups were placed in a side chamber and a novel object placed in the other and the time spent in proximity of these across a 10-minute span was recorded.

Several factors can influence parenting behaviour indirectly such as litter size, parent motility, coping with novelty, and olfaction. We saw no significant differences in litter size recorded

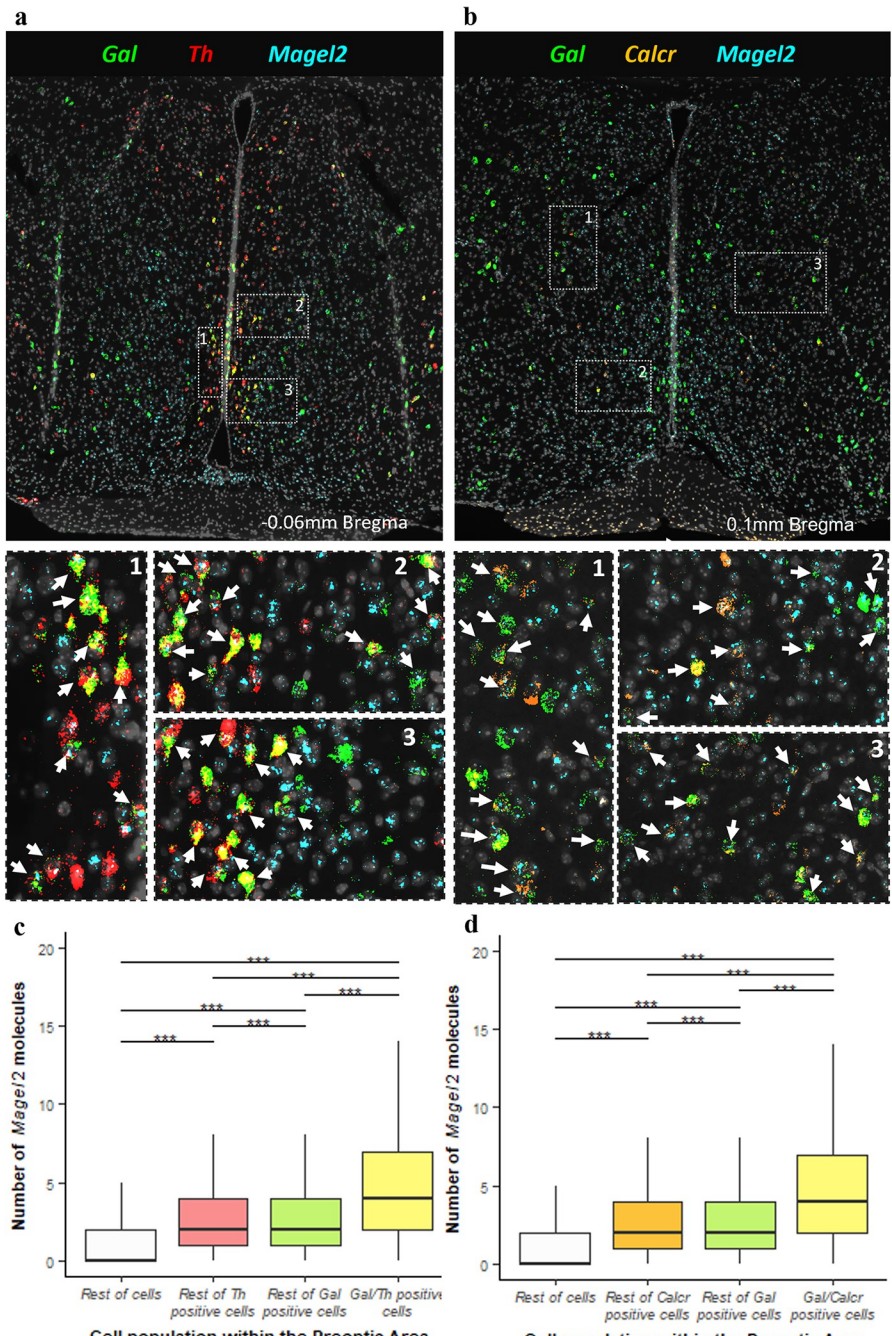

**Fig 2. In situ coexpression of *Magel2* in the POA. (A)** ***Top*** Low magnification image of hypothalamic section after *in situ* amplification of *Gal* (green), *Th* (red), and *Magel2* (turquoise). ***Bottom*** High-resolution image of three open white dashed boxes numbered 1–3. Examples of co-expression of *Gal*, *Th* and *Magel2* in one cell are indicated with white arrows. **(B)** ***Top*** Low magnification image of hypothalamic section after *in situ* amplification of *Gal* (green), *Calcr* (orange), and *Magel2* (turquoise). ***Bottom*** High-resolution image of the three open white dashed boxes numbered 1–3 from the top image. Examples of co-expression of *Gal*, *Calcr* and *Magel2* in one cell are indicated with white arrows. **(C)** Number of *Magel2* RNA molecules detected in different cell types from all sections. *Gal/Th* cells expressed significantly more RNA molecules of *Magel2* than the other cell types, even including *Gal* expressing and *Th* expressing cells separately (H [3] = 5313.6, p = 2.2x10-16, ***P<0.001, post hoc Dunn test). **(D)** Number of *Magel2* RNA molecules detected in different cell types from all sections. *Gal/Calcr* cells expressed significantly more RNA molecules of *Magel2* than the other cell types, even including *Gal* expressing and Calcr expressing cells separately (H [3] = 17152, p = 2.2x10-16, ***P<0.001, post hoc Dunn test).

at P2 (Fig BA in S1 Text) between the cohorts of WTs and *Magel2*-nulls. There were no overt motility disadvantages between the *Magel2*-null and WT individuals in each group (mothers, fathers, virgins), with no significant differences in velocity in the retrieval task (Fig BB in S1 Text), and no differences in number of times moving between the chambers in the three chambers assessment (Fig BC in S1 Text). There were also no significant differences in time taken to first sniff and investigate the pups (Fig BD in S1 Text) indicating no overt olfactory deficit. Finally, to reduce the novelty aspect of these tests, the test was performed in the home cage with the home nest and their own pups, and animals were thoroughly habituated to the apparatus in the preceding days.

## Magel2-null mothers displayed poorer nest building and less pup-directed motivation

The three maternal cohorts (summarised in Fig 3A) were as follows: **WT(WT)**—WT female paired with WT male, mothering WT pups, **WT(*Magel2*)**—WT female paired with mutant *Magel2*-null($^{+/p}$) male, mothering WT and mutant pups (*Magel2*-null($^{+/p}$)) and **Magel2-null**—mutant *Magel2*-null($^{+/p}$) female paired with WT male, mothering WT and functionally WT pups (*Magel2*$^{m/+}$)

Success rate in the task (retrieve 3 pups and rebuild the nest) differed between the three maternal cohorts (Fig 3B and 3H [2] = 20.86, $p$ = 2.95 x $10^{-5}$). *Magel2*-null mothers paired with WT males displayed a significantly worse performance during the retrieval-nest-building task than both WT(WT) ($p$ = 0.0004) and WT(*Magel2*) ($p$ = 0.0003). Both WT maternal cohorts successfully retrieved all 3 pups and rebuilt their nest in the one-hour time frame whereas only 56% of *Magel2*-null mothers achieved the same. The time taken to complete the task differed between the maternal cohorts (F(2, 62) = 21.48, $p$ = 8.16x$10^{-8}$) with both WT (WT) ($p$ = 1.9x$10^{-6}$) and WT(*Magel2*) ($p$ = 1.6x$10^{-6}$) completing the task faster than the *Magel2*-null mothers (Fig 3C).

During the retrieval-nest-building task, there were no significant differences in time taken to retrieve the first pup (Fig 3D; F(2, 62) = 1.35, $p$ = 0.271) and final pup (Fig 3E; F(2, 62) = 1.98, $p$ = 0.07), indicating that *Magel2*-null mothers have comparable retrieval ability to their WT comparisons. Indeed, 100% of the *Magel2*-null and WT mothers successfully retrieved the three pups to the nest area (Fig 3F). However, 46% of *Magel2*-null mothers failed to build a suitable quality nest and the maternal cohorts differed in both the time taken to rebuild the home nest to a Level 3 state (Fig 3G; F(2, 62) = 21.48, $p$ = 8.16x$10^{-8}$) and the final quality of the rebuilt nest (Fig 3H, H [2] = 20.06, $p$ = 4.40 x $10^{-5}$). *Magel2*-null mothers were slower to build a level 3 nest than WT(WT) ($p$ = 1.90x$10^{-6}$) and WT(*Magel2*) ($p$ = 1.6x$10^{-6}$) and had significantly poorer quality nests than WT(WT) ($p$ = 0.0003) and WT(*Magel2*) ($p$ = 0.0007).

In addition to difference in nest building, there was a difference between the maternal cohorts in the proportion of time that mothers spent in pup-directed behaviour (PDB) up until that successful final retrieval (Fig 3I; F(2, 62) = 7.12, $p$ = 0.002). *Magel2*-null mothers spent a significantly smaller proportion of their time leading up to the successful final retrieval engaging in PDB compared to WT(WT) ($p$ = 0.0035) and WT(*Magel2*) ($p$ = 0.013). This difference was also found when considering the proportion of time spent in pup-directed behaviour until the task was finished (either upon completion of the task or upon the expiration of the one-hour testing time; Fig 3J; F(2, 62) = 21.02, $p$ = 1.07x$10^{-7}$) with *Magel2*-null mothers spending a smaller proportion of their time compared to WT(WT) ($p$ = 18.20x$10^{-7}$) and WT (*Magel2*) ($p$ = 6.80x$10^{-6}$). The three chambers assessment was used as a second independent measure of pup affiliation and parental motivation and WT(WT) ($t$ [17] = 2.15, $p$ = 0.045) and WT(*Magel2*) ($t$ [20] = 2.37, $p$ = 0.028) both spent significantly more time in vicinity of the

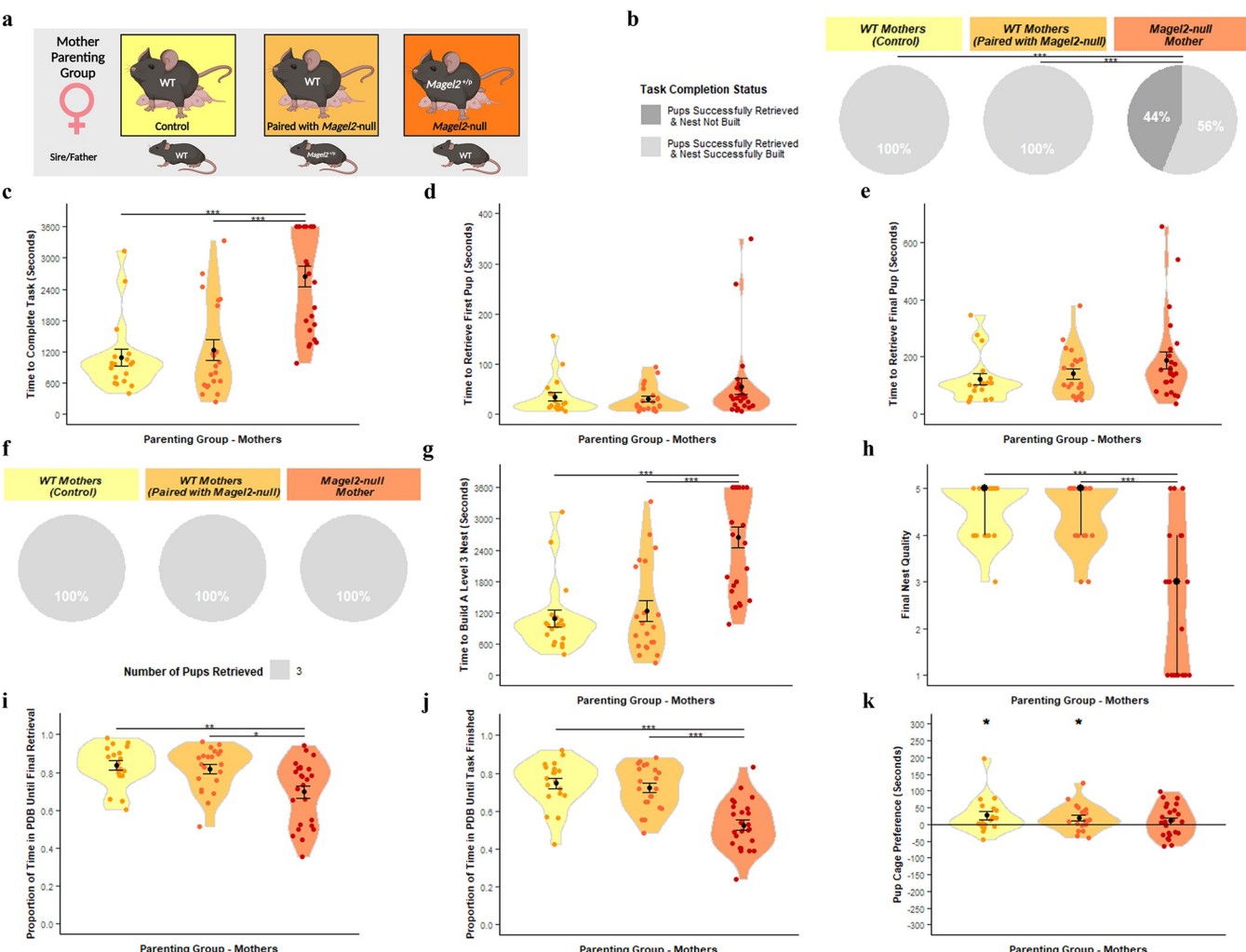

**Fig 3. Mother Parenting Assessment.** (A) Schematic of behavioural paradigm with mothers. WT (Paired with WT) *n* = 19, WT (Paired with *Magel2*-null male) *n* = 21, *Magel2*-null *n* = 25 (B) Task Completion Status at conclusion of Retrieval/Nest Building Task, for a visual aid of the task, see Fig 8C. Mothers were categorised on their ability to rebuild their nest to a level 3 quality and to retrieve the pups into the nest within the one-hour time limit. Percentages of mothers falling within those categories are shown. (C) Time taken to complete the Retrieval/Nest Building Task. Time taken to retrieve all three pups to the area where the nest was rebuilt and to rebuild the nest to a level 3 quality or higher. (D) Time taken to retrieve the first pup to the nest in the Retrieval/Nest Building Task. (E) Time taken to retrieve the final/third pup to the nest in the Retrieval/Nest Building Task and hence completing the retrieval portion of the task. (F) Number of pups retrieved at conclusion of Retrieval/Nest Building Task. Mothers were categorised on the number of pups they successfully retrieved and percentages of mothers falling within those categories are shown. (G) Time taken to re-build the nest within the Retrieval/Nest Building task to a level 3 quality or higher. Time was recorded for when the nest being constructed by the mothers scored a level 3 quality score (the point when the nest takes functional shape). (H) Nest Quality score at conclusion of Retrieval/Nest Building Task. Mother's rebuilt nests were scored from 0–5 upon completion of the test. (I) Proportion of time spent engaged in pup-directed behaviour (PDB) until the final/third pup was retrieved to the nest/until task time expires in the Retrieval/Nest Building Task. Mother's behaviours were scored continuously through the one-hour trial. Pup-directed behaviour included time spent engaging in licking, grooming, sniffing, retrieving pups, alongside nest building and crouching in nest (only while pups were present in the nest). (J) Proportion of time spent engaged in PDB until the final/third pup was retrieved to the nest and the nest was rebuilt to a level 3 standard/until task time expires in the Retrieval/Nest Building Task. (K) Pup preference score in Three Chambers Assessment. Pup preference scores was calculated as time the mother spent within a 15cm zone around the pups minus time spent within a 15cm zone around the novel object. Positive values indicate a preference for proximity to pups. Significance for continuous variables determined using one-way ANOVA and Bonferroni-corrected pairwise t tests. Significance for categorical variables determined using Kruskal-Wallis test and Bonferroni-corrected Dunn test. Statistical significance: $^*p < 0.05$, $^{**}p < 0.01$, and $^{***}p < 0.001$. Created with Biorender.

pups than the novel object and hence demonstrated a pup-preference (Fig 3K). *Magel2*-null mothers did not demonstrate a significant pup-preference score ($t$ [24] = 1.07, $p$ = 0.29). In summary, *Magel2*-null mothers do not show retrieval deficits but do show a general reduction

in pup-directed motivation and deficits in nest building indicating parenting behaviour was insulted.

## Magel2-null fathers performed poorly on all measures of parenting behaviour

The prerequisite for inducing parenting behaviour in murine fathers is an extended cohabitation phase post-coitus with a pregnant female, and so all males used were permanently co-housed with the females. The three paternal cohorts were produced from the same pairing as the maternal cohorts (summarised in Fig 4A) and were as follows: **WT(WT)**—WT male paired with WT female fathering WT pups, **WT(*Magel2*)**—WT male paired with mutant *Magel2*-null$^{(+/p)}$ female fathering WT and functionally WT pups (*Magel2*$^{m/+}$) and **_Magel2_-null**-mutant *Magel2*-null$^{(+/p)}$ male paired with WT female fathering WT and mutant pups (*Magel2-null*$^{(+/p)}$).

Success rate in the task (retrieve 3 pups and rebuild the nest) differed between the three paternal cohort (Fig 4B; H [2] = 26.86, *p* = 1.47 x 10$^{-6}$). *Magel2*-null fathers displayed a significantly worse performance during the retrieval-nest-building task than both WT(WT) (*p* = 5.00x10$^{-6}$) and WT(*Magel2*) (*p* = 0.0001). 79% of WT(WT) completed the test successfully, 60% of WT(*Magel2*) and only 9% of *Magel2*-null fathers successfully completed the task. All paternal cohorts had a percentage of failures, but the time taken to complete the task differed between the paternal cohorts (F(2, 63) = 13.24, *p* = 1.59x10$^{-5}$) with *Magel2*-null fathers completing the task slower than both WT(WT) (*p* = 2.00x10$^{-5}$) and WT(*Magel2*) (*p* = 0.001). (Fig 4C).

Within the retrieval component, there were significant differences between the paternal cohorts in the time taken to retrieve both the first pup (Fig 4D; F(2, 63) = 11.52, *p* = 5.44x10$^{-5}$) and last pup (Fig 4E; F(2, 63) = 12.86, *p* = 4.59x10$^{-5}$). *Magel2*-null fathers were significantly slower to retrieve the first pup compared to WT(WT) (*p* = 0.0001) and WT(*Magel2*) (*p* = 0.001), and the final pup compared to WT(WT) (*p* = 0.0001) and WT(*Magel2*) (*p* = 0.0005). This manifested in differences in the number of pups they retrieved by the end of the task (Fig 4F; H [2] = 23.06, *p* = 9.83 x 10$^{-6}$). *Magel2*-null fathers retrieved significantly fewer pups than the WT(WT) (*p* = 8.30x10-5) and WT(*Magel2*) (*p* = 0.0001). 64% of *Magel2*-null fathers failed to retrieve any pups while only 20% and 16% of WT fathers failed to retrieve no pups. Within the nest building component, the paternal cohorts differed in both the time taken to rebuild the home nest to a Level 3 state (Fig 4G; F(2, 63) = 18.72, *p* = 4.15x10$^{-7}$) and the final quality of the rebuilt nest (Fig 4H, H [2] = 12.02, *p* = 1.47 x 10$^{-6}$). *Magel2*-null fathers were slower to build a level 3 nest than WT(WT) (*p* = 3.60x10$^{-7}$) and WT(*Magel2*) (*p* = 3.20x10$^{-4}$) and had significantly poorer quality nests than WT(WT) (*p* = 0.002) but not WT(*Magel2*) (*p* = 0.66).

There were differences between the paternal cohorts in the proportion of the time devoted to pup-directed behaviour (PDB) until the final pup was retrieved (Fig 4I; F(2, 63) = 5.90, *p* = 0.004) and until the task was finished/one hour expired (Fig 4J; F(2, 63) = 9.52, *p* = 0.0002). *Magel2*-null fathers dedicated a smaller proportion of their time to PDB than the WT(WT) until final retrieval (*p* = 0.006) and until task finished (*p* = 0.031) and, than the WT(*Magel2*) until final retrieval (*p* = 0.0002) and until task finished (*p* = 0.011). None of the fathers demonstrated a significant pup-preference score in the three chambers test (Fig 4K). However, *Magel2*-null fathers (*t* [20] = -3.18, *p* = 0.005) and WT(*Magel2*) fathers (*t* [24] = -2.58, p = 0.016) demonstrated a significant preference for the object zone compared to the pup zone, so a significant pup-avoidance score. In summary, *Magel2*-null fathers showed parental deficits in all metrics assessed here.

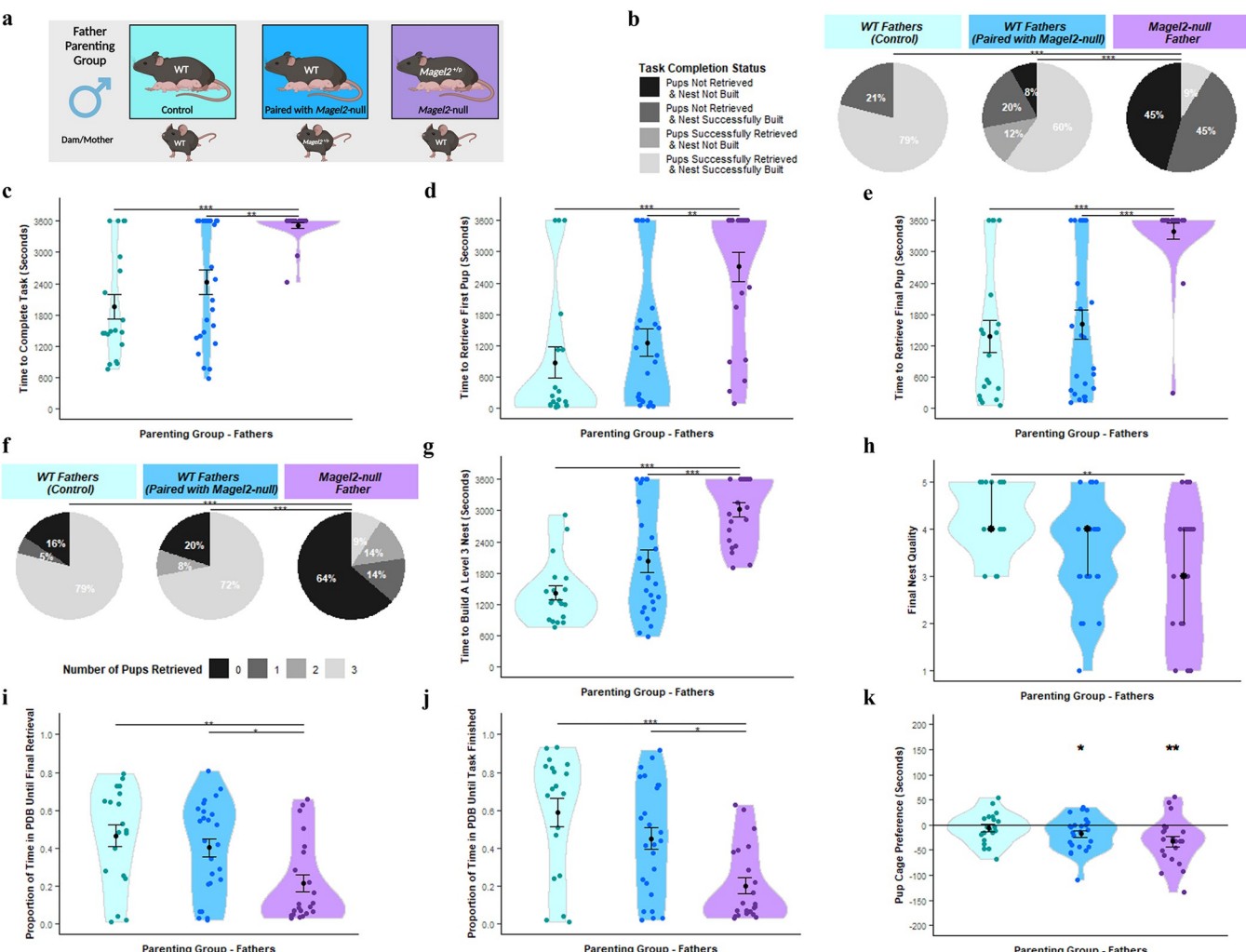

**Fig 4. Father Parenting Assessment.** (A) Schematic of behavioural paradigm with fathers. WT (Paired with WT) *n = 19*, WT (Paired with *Magel2*-null female) *n = 25*, *Magel2*-null *n = 22* (B) Task Completion Status at conclusion of Retrieval/Nest Building Task, for a visual aid of the task, see Fig 8C. Fathers were categorised on their ability to rebuild their nest to a level 3 quality and to retrieve the pups into the nest within the one-hour time limit. Percentages of fathers falling within those categories are shown. (C) Time taken to complete the Retrieval/Nest Building Task. Time taken to retrieve all three pups to the area where the nest was rebuilt and to rebuild the nest to a level 3 quality or higher. (D) Time taken to retrieve the first pup to the nest in the Retrieval/Nest Building Task. (E) Time taken to retrieve the final/third pup to the nest in the Retrieval/Nest Building Task and hence completing the retrieval portion of the task. (F) Number of pups retrieved at conclusion of Retrieval/Nest Building Task. Fathers were categorised on the number of pups they successfully retrieved and percentages of fathers falling within those categories are shown. (G) Time taken to re-build the nest within the Retrieval/Nest Building task to a level 3 quality or higher. Time was recorded for when the nest being constructed by the fathers scored a level 3 quality score (the point when the nest takes functional shape). (H) Nest Quality score at conclusion of Retrieval/Nest Building Task. Father's rebuilt nests were scored from 0–5 upon completion of the test. (I) Proportion of time spent engaged in pup-directed behaviour (PDB) until the final/third pup was retrieved to the nest/until task time expires in the Retrieval/Nest Building Task. Father's behaviours were scored continuously through the one-hour trial. Pup-directed behaviour included time spent engaging in licking, grooming, sniffing, retrieving pups, alongside nest building and crouching in nest (only while pups were present in the nest). (J) Proportion of time spent engaged in PDB until the final/third pup was retrieved to the nest and the nest was rebuilt to a level 3 standard/until task time expires in the Retrieval/Nest Building Task. (K) Pup preference score in Three Chambers Assessment. Pup preference scores was calculated as time the father spent within a 15cm zone around the pups minus time spent within a 15cm zone around the novel object. Positive values indicate a preference for proximity to pups. Significance for continuous variables determined using one-way ANOVA and Bonferroni-corrected pairwise t tests. Significance for categorical variables determined using Kruskal-Wallis test and Bonferroni-corrected Dunn test. Statistical significance: *$p < 0.05$, **$p < 0.01$, and ***$p < 0.001$. Created with Biorender.

## Magel2-null pups had no significant effect on task completion

It has already been suggested that pups inheriting the paternally mutated allele of *Magel2* have behavioural differences that can influence maternal preference during retrieval [36].

Additionally, *Magel2*-null pups are known to have growth deficits [37]. Hence, in an attempt to minimize the number of *Magel2*-null pups that were represented in the test litters of *Magel2*-null fathers and the paired WT(*Magel2*) mothers, we weighed all pups at P2 and marked the three heaviest as the test pups. This was done for all maternal and paternal cohorts. Genotyping of all assessment pups showed that 45% of the pups from *Magel2*-null mothers paired with WT males were *Magel2*$^{(m/+)}$ (functionally WT) while only 24% of the pups from *Magel2*-null fathers paired with WT females were *Magel2*-null$^{(+/p)}$ (functionally mutant) which meant that 50% of the *Magel2*-null fathers and their associated WT mothers had no mutant *Magel2*-null pups in their test litters.

However, 11/22 of the *Magel2*-null fathers and their associated WT mothers still had at least one mutant pup in their assessment. To assess whether mutant pups were influencing the outcome of our assessment, we first compared the average retrieval times for a mutant pup in these 11 litters relative to WT littermates (see Fig 5) and found no significant differences in the time for WT(*Magel2*) mothers to retrieve a mutant pup compared to WT pups (W = 117, *p* = 0.53, Wilcoxon Ranked-Sum Test). The same was seen for *Magel2*-null fathers (W = 117, *p* = 0.43, Wilcoxon Ranked-Sum Test). Secondly, we re-ran the analyses of the previous sections (mothers and fathers) while excluding data from litters containing mutant pups and produced the same statistical findings, further suggesting that the presence of *Magel2*-null pups was not influencing the parental behaviour we observed at P3-P5. (See Table F in S1 Text).

## Magel2-null virgin females displayed poorer retrieval behaviour and less pup-directed motivation

Virgin females display parenting behaviour spontaneously, albeit less reliably than mothers. However, subsequent exposures to pups improves the likely manifestation of parenting behaviour [38, 39]. To incorporate this improvement effect, virgin females underwent two retrieval-nest-building tests on subsequent days before performing the three chambers test on the following day. Cohorts consisted of **WT** and ***Magel2*-null**$^{(+/p)}$ females, tested with a unique set of three WT pups derived from WT x WT pairings (See Fig 6A).

For success rate in the task (Fig 6B), there was a main effect of Genotype (H [1] = 12.36, *p* = 4.39 x 10$^{-4}$) with *Magel2*-null virgin females displaying a significantly worse performance during the hybrid-retrieval task than the WTs and a main effect of Exposure (H [1] = 12.80, *p* = 3.46 x 10$^{-4}$) with a higher task success in the second exposure. The WT success rate was 75% on first exposure and 90% on the second exposure, while for *Magel2*-null virgin females they had a 30% success rate followed by a 65% success rate. For time to complete the task (Fig 6C), there was a significant interaction effect (F [1,38] = 6.22, *p* = 0.017) and simple main effects analysis revealed that *Magel2*-null virgin females took longer to finish the task compared to WT virgin females only in the first exposure (p = 0.0078) but not in the second. *Magel2*-null virgin females also saw significant improvement between the first and second exposure (p = 0.0064) while the WT virgin females did not.

Focusing on the retrieval component, Fig 6D and 6E display the time taken to retrieve the first and last pups respectively. For the time to retrieve the first pup, there was a significant interaction effect (F [1,38] = 4.995, *p* = 0.031) and simple main effects analysis revealed that *Magel2*-null virgin females took longer to retrieve the first pup in both the first exposure (p = 0.00004) and in the second exposure (p = 0.025). Neither *Magel2*-null nor WT virgin females saw significant improvement upon second exposure. For the time to retrieve the final pup, there was a significant interaction effect (F [1,38] = 4.828, *p* = 0.034) and simple main effects analysis revealed that *Magel2*-null virgin females took longer to retrieve the first pup in the first exposure (p = 0.00027) but not in the second (p = 0.071). *Magel2*-null virgin females

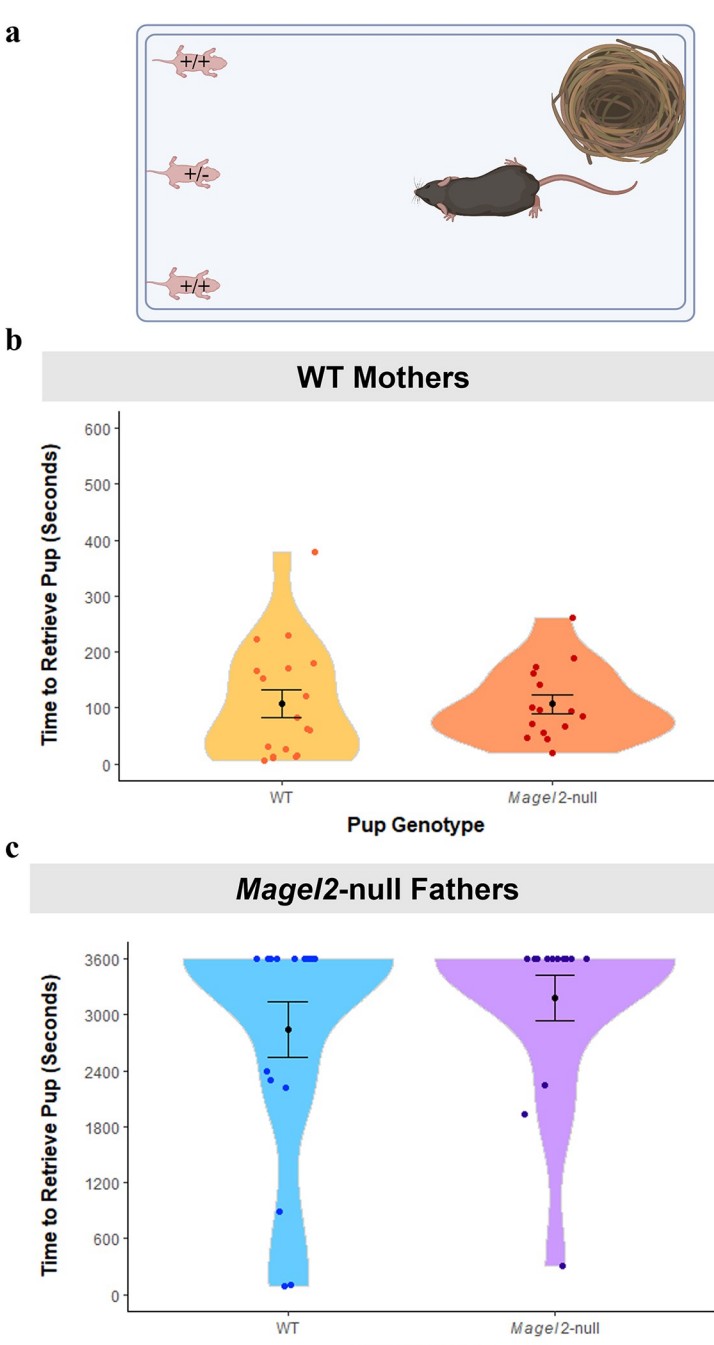

**Fig 5. Pup retrieval times for WT (n = 18) and *Magel2*-null pups (n = 15) within mixed genotype litter retrievals (n = 11).** Mixed litters could only result from pairings with WT females and *Magel2*-null males. (A) Schematic showing retrieval set up with a mutant pup present as one of the three animals to be retrieved. Animals had a maximum of 3600 seconds to retrieve pups. (B) Time to retrieve WT and mutant pups for WT mothers (C) Time to retrieve WT and mutant pups for *Magel2*-null fathers. Created with Biorender.

also saw significant improvement between the first and second exposure (p = 0.0086) while the WT virgin females did not. For the number of pups retrieved by the end of the trail (Fig 6F), there was a significant main effect of Genotype (H [1] = 9.62, *p* = 0.0019) with *Magel2*-null

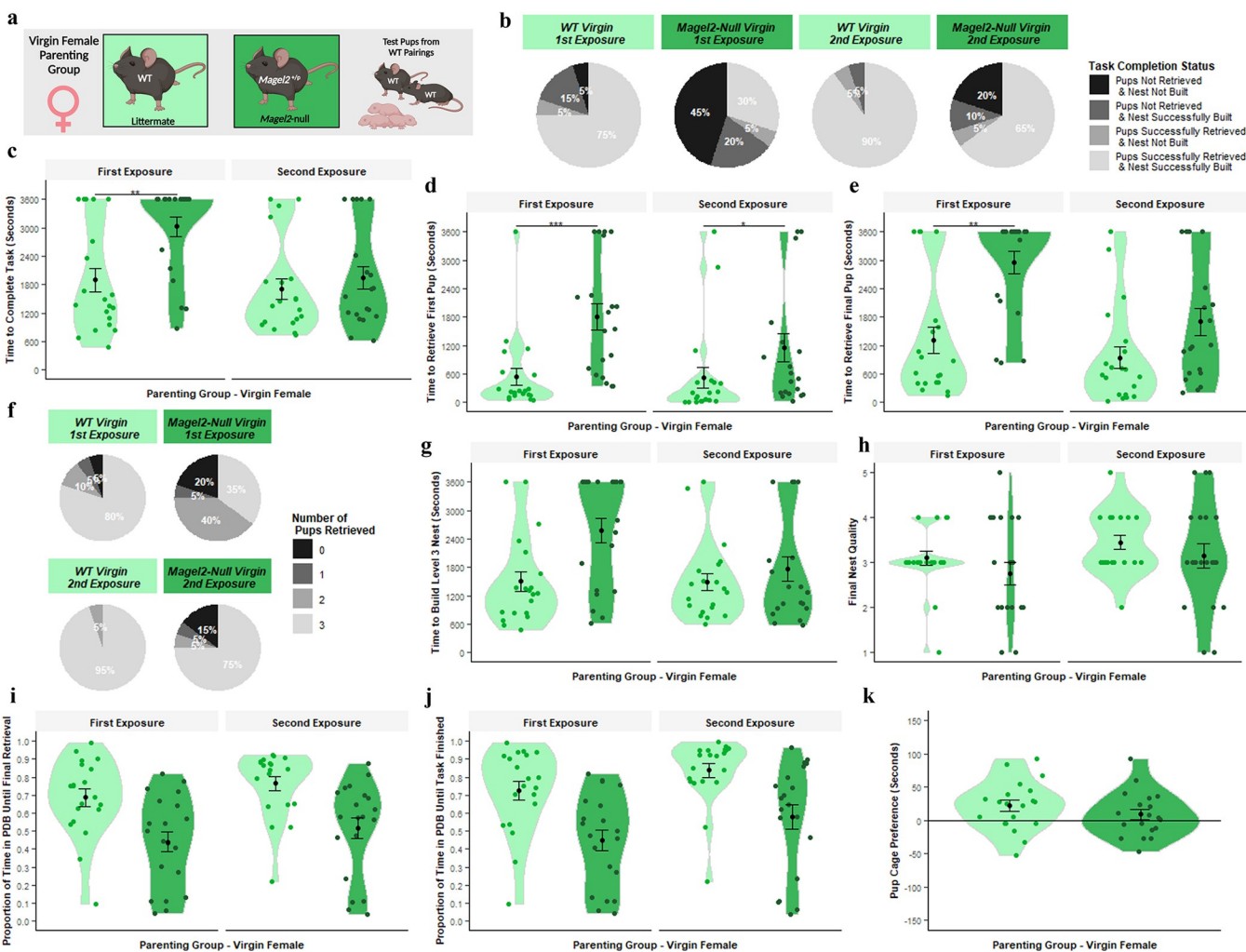

**Fig 6. Virgin Female Parenting Assessment.** (A) Schematic of behavioural paradigm with virgin females. WT (Littermate) *n = 20*, *Magel2*-null *n = 20*. Each female was tested with 3 unique pups acquired from WT x WT pairings. The Retrieval/Nest Building Task was carried out twice for each female (First Exposure and Second Exposure), for a visual aid of the task, see Fig 8C. (B) Task Completion Status at conclusion of the first and second Retrieval/Nest Building Task. Virgin females were categorised on their ability to rebuild their nest to a level 3 quality and to retrieve the pups into the nest within the one-hour time limit. Percentages of virgin females falling within those categories are shown. (C) Time taken to complete the Retrieval/Nest Building Task. Time taken to retrieve all three pups to the area where the nest was rebuilt and to rebuild the nest to a level 3 quality or higher. (D) Time taken to retrieve the first pup to the nest in the Retrieval/Nest Building Task. (E) Time taken to retrieve the final/third pup to the nest in the Retrieval/Nest Building Task and hence completing the retrieval portion of the task. (F) Number of pups retrieved at conclusion of Retrieval/Nest Building Task. Virgin females were categorised on the number of pups they successfully retrieved and percentages of Virgin females falling within those categories are shown. (G) Time taken to re-build the nest within the Retrieval/Nest Building task to a level 3 quality or higher. Time was recorded for when the nest being constructed by the virgin females scored a level 3 quality score (the point when the nest takes functional shape). (H) Nest Quality score at conclusion of Retrieval/Nest Building Task. Virgin female's rebuilt nests were scored from 0–5 upon completion of the test. (I) Proportion of time spent engaged in pup-directed behaviour (PDB) until the final/third pup was retrieved to the nest/until task time expires in the Retrieval/Nest Building Task. Virgin female's behaviours were scored continuously through the one-hour trial. Pup-directed behaviour included time spent engaging in licking, grooming, sniffing, retrieving pups, alongside nest building and crouching in nest (only while pups were present in the nest). (J) Proportion of time spent engaged in PDB until the final/third pup was retrieved to the nest and the nest was rebuilt to a level 3 standard/until task time expires in the Retrieval/Nest Building Task. (K) Pup preference score in Three Chambers Assessment. Pup preference scores was calculated as time the virgin female spent within a 15cm zone around the pups minus time spent within a 15cm zone around the novel object. Positive values indicate a preference for proximity to pups. Significance for continuous variables determined using two-way ANOVA and Bonferroni-corrected pairwise t tests. Significance for categorical variables determined using Kruskal-Wallis test and Bonferroni-corrected Dunn test. Statistical significance: *$p < 0.05$, **$p < 0.01$, and ***$p < 0.001$. Created with Biorender.

virgin females having retrieved fewer pups than WTs. There was a main effect of Exposure (H [1] = 11.68, *p* = 0.0006) with more pups retrieved in the second exposure. 80% of WT virgin females retrieve all 3 pups in the first exposure compared to 35% of *Magel2*-null virgin females.

In the nest building component, there was a main effect of Genotype for the time taken to construct a Level 3 nest (Fig 6G; F [1,38] = 6.76, *p* = 0.013) with WTs building level 3 nests faster, as well as a main effect of Exposure (F [1,38] = 4.41, *p* = 0.043) with level 3 nests built faster in the second exposure. For nest quality at the end of the assessment (Fig 6H), there was a main effect of Exposure (H [1] = 4.61, *p* = 0.032) with higher quality nests built in the second exposure but no main effect of Genotype, which indicates that while virgin mice of both genotypes do not tend to build high quality nests within the hour, the WTs are still quicker to build a suitable nest for the pups. When considering pup-directed behaviour (PDB), there was a main effect of Genotype for the PDB up to final retrieval (Fig 6I; F [1,38] = 15.28, *p* = 0.0004) and for the PDB until task finished (Fig 6J; F [1,38] = 15.90, *p* = 0.0003),with *Magel2*-null virgin females dedicating a smaller proportion of their time to PDB than the WTs. There was a main effect of Exposure for the PDB until retrieval (F [1,38] = 4.311, *p* = 0.045) and task finished (F [1,38] = 11.14, *p* = 0.002) with more PDB displayed in the second exposure. Additionally, the WT virgin females (following their two exposures) demonstrated a significant pup-preference score during the three chambers assessment (Fig 6K; *t* [19] = 2.70, *p* = 0.014). This was not true for the *Magel2*-null virgin females who failed to show a preference for either the pups or the novel object (*t* [18] = 1.15, *p* = 0.26). In summary, *Magel2*-null virgin females showed a significant reduction in innate interest in pups and had poorer retrieval behaviour than WT comparisons. Both groups improved upon a second pup exposure but the differences between genotypes remained.

## Magel2-null mothers and fathers have reduced c-Fos activity in the POA upon exposure to pups partially explained by a reduction in Gal/Calcr expressing cells

To assess the impact of a loss of *Magel2* on neuronal activity in the POA we assessed expression of *c-Fos* alongside *Gal/Calcr* using RNAscope. This was performed in the POA of *Magel2*-null and WT Mothers and Fathers following exposure to pups (after a period of isolation) vs. male and female non-exposed controls. Fig 7A displays a representative POA image from pup-exposed *Magel2*-null and WT Mothers. POA morphology and total cell count (Fig CA in S1 Text) were comparable between Genotypes (F [1,27] = 0.122, *p* = 0.73) suggesting loss of *Magel2* was not affecting the gross cellular composition of the POA. For all subsequent RNA counts we normalised counts to number of RNA counts per 1000 POA cells.

For *c-Fos*, we saw that Pup-Exposed mice had 51.7% more *c-Fos* RNA produced in the POA upon exposure to pups compared to controls (F [1,23] = 97.91, *p* = $2.24 \times 10^{-6}$). A main effect of Genotype was also seen; *Magel2*-null mice had 13.8% fewer *c-Fos* positive cells (F [1,23] = 5.006, *p* = 0.035, Fig 7B) and a 12% reduction in general *c-Fos* RNA in the POA (F [1,23] = 4.73, *p* = 0.04, Fig CB in S1 Text). This was true for *Magel2*-null males compared to WT males and *Magel2*-null females compared to WT females. For comparison, we did not observe a main effect of Genotype in *c-Fos* expression in cortical cells suggesting the *c-Fos* difference we see in the POA is specific (Fig CC in S1 Text). *Gal* and *Calcr* RNA molecules were quantified alongside *c-Fos* RNA, and we saw that there were 100.3% more *c-Fos* positive *Gal/Calcr* cells when exposing mice to pups (F [1,23] = 98.39, *p* = $8.91 \times 10^{-10}$) and 28.4% more *c-Fos* positive *Gal* cells (F [1,23] = 45.60, p = $6.9 \times 10^{-7}$). There was also a main effect of Genotype, showing that *Magel2*-null mice had a 23.3% reduction in *c-Fos* positive *Gal* cells (F [1,23] = 10.43,

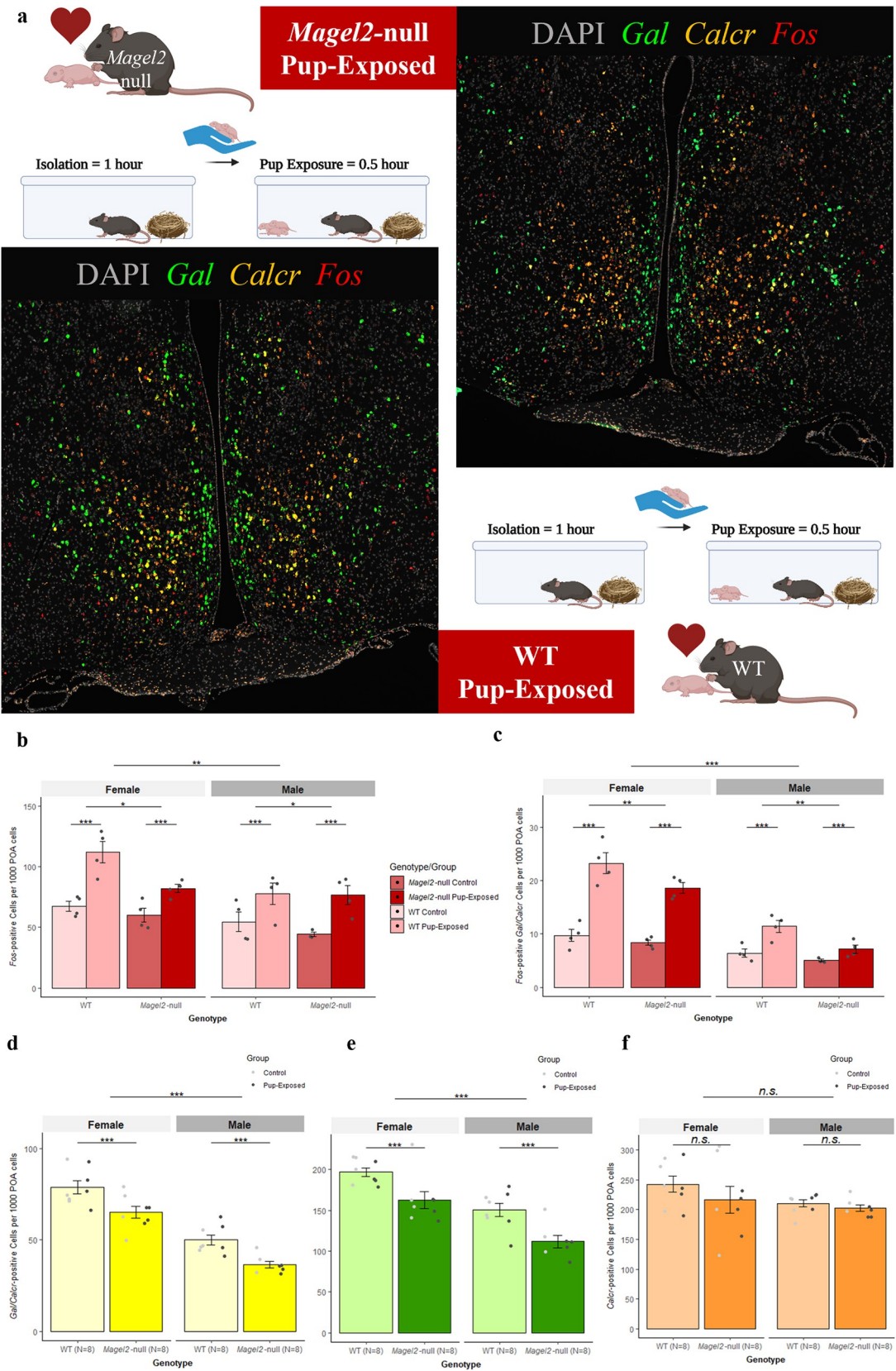

**Fig 7. *c-Fos* Expression in the POA of Pup-Exposed and WT mice.** (A) Representative POA *Gal/Calcr c-Fos* Images from Pup-Exposed mice. *Magel2*-null mice (*Top*) and WT mice (*Bottom*) were either paired to produce litters and then used as the Pup-Exposed group (N = 4 per genotype) or were left undisturbed to act as Controls (N = 4 per genotype). Pup exposure consisted of reintroducing pups to the mice following a 1-hour isolation period and exposing the mice to pups for 30 minutes prior to tissue harvest. Control mice were isolated for 1 hour but were not exposed to pups and underwent tissue harvest immediately afterwards away. Images present DAPI (Grey) stained nuclei alongside RNA molecules of *Gal* (Green), *Calcr* (Orange) and *c-Fos* (Red). (B) Number of *c-Fos* positive ($\geq$5 molecules) cells per 1000 POA cells of *Magel2*-null mice and WT mice either exposed to pups or controls. (C) Number of *c-Fos* positive (5+ molecules) cells also expressing *Gal* and *Calcr* (2+ molecules) per 1000 POA cells of *Magel2*-null mice and WT mice either exposed to pups or controls. (D) Number of *Gal/Calcr* positive (2 + molecules each) cells per 1000 POA cells of *Magel2*-null mice and WT (regardless of exposure). (E) Number of *Gal* positive (2 + molecules) cells per 1000 POA cells of *Magel2*-null mice and WT mice (regardless of exposure). (F) Number of *Calcr* positive (2+ molecules) cells per 1000 POA cells of *Magel2*-null mice and WT mice (regardless of exposure). Created with Biorender.

$p$ = 0.0037, Fig CD in S1 Text) and 20.4% fewer *c-Fos* positive *Gal/Calcr* cells (F [1,23] = 11.25, p = 0.0028, Fig 7C), but no effect of Genotype for *c-Fos* positive *Calcr* cells (F [1,23] = 4.01, $p$ = 0.0503, Fig CE in S1 Text). Again, these differences were true for *Magel2*-null males compared to WT males and *Magel2*-null females compared to WT females. There were no differences in average number of *c-Fos* molecules per cell between *Magel2*-null and WT (F [1,23] = 1.16, $p$ = 0.21) suggesting that there was a loss of *c-Fos* positive cells rather than a loss of *c-Fos* expression within positive cells (Fig CF in S1 Text).

A simple explanation for the loss of *c-Fos* positive cells but no broad loss of POA cells (Fig CA in S1 Text) would be a specific loss of highly active cell types (such as the *Gal/Calcr* cells) in *Magel2*-null animals. When comparing the expression of *Gal* and *Calcr* between *Magel2*-null and WT, we saw, a 19.6% reduction in *Gal/Calcr* expressing cells (F [1,23] = 16.04, $p$ = 0.0006, Fig 7D), a 19.9% reduction in *Gal* expressing cells (F [1,23] = 17.54, $p$ = 0.0004, Fig 7E) and a 20.3% reduction in *Gal* RNA molecules (F [1,23] = 11.708, $p$ = 0.0023, Fig DA in S1 Text) in *Magel2*-null mutant females compared to WT females and in mutant males compared to WT females. There was no significant reduction in *Calcr* cells (F [1,23] = 1.22, $p$ = 0.28, Fig 7F) or *Calcr* RNA molecules (F [1,23] = 1.14, $p$ = 0.30 Fig DB in S1 Text), indicating that *Magel2*-null mice have a specific reduction in *Gal* expression in the POA. When we normalize the number of *c-Fos* positive *Gal* and *Gal/Calcr* cells by the total number of *Gal* and *Gal/Calcr* cells, we find that the significant reduction in *c-Fos in Gal* and *Gal/Calcr* cells in *Magel2*-null mice is lost (Fig DC in S1 Text), indicating that the *Gal* and *Gal/Calcr* cells of *Magel2*-null mice are just as neurologically active as the WT, but the *Magel2*-null mice simply have fewer of these parenting-associated *Gal/Calcr* cells than their WT comparisons.

Despite the previous differences observed for both males and females, main effects of Sex were seen in number of *Gal* (F [1,23] = 33.73, $p$ = $6.45 \times 10^{-6}$), *Gal/Calcr* (F [1,23] = 83.04, $p$ = $4.28 \times 10^{-9}$) and *c-Fos* (F [1,23] = 12.67, $p$ = 0.002) positive cells with females possessing more of these parenting-associated cell types and a larger *c-Fos* response than males. Outputs of all ANOVAs performed from this experiment can be found in S2 Table.

## Discussion

In this study, we used a systems biology approach to assess whether genomic imprinting is important for mediating parental care by examining whether imprinted gene expression is enriched in the neural circuitry that controls these behaviours. Using publicly available single cell RNA sequencing data we demonstrated that imprinted gene expression is enriched in the parenting-associated galanin neurons of the POA [40, 41]. We then tested the validity of inferring function from expression by focusing on *Magel2*. We confirmed the elevated expression of *Magel2* in parenting-associated POA neurons and then assessed the parenting behaviour in *Magel2*-null mothers, fathers and virgin females. We found overlapping deficits in parenting

performance and motivation in all three groups highlighting the fundamental importance of *Magel2* for parenting behaviour. Furthermore, we found that *Magel2* deletion impacts the POA directly since *Magel2*-null females and males had a less significant c-Fos response in the POA upon exposure to pups compared to WT's and a general reduction in *Gal* expression in the POA. Taken together with our previous findings [30], this systematic investigation indicates that parental care is a key brain function upon which imprinted genes converge.

We provide evidence, through enrichment of expression, that regulation of parental behaviour is a key functional output of imprinted genes. Namely we saw an enrichment of imprinted genes in a specific population of POA galanin neurons that are master regulators of parenting behaviour [27, 28]. 21 imprinted genes (1/6 of the genes analysed) were significantly expressed in neurons expressing the parenting associated markers–*Gal/Th/Calcr/Brs3*. Of these, *Peg3*, *Dio3* and *Mest*, were previously associated with maternal behaviour deficits when deleted in female mice. A fourth gene associated with maternal behaviour, *Peg13*, was not sequenced in the datasets we analysed. The gene we chose to assess, *Magel2*, had not been previously linked with parental care provision. *Magel2* displayed > 17-fold higher expression in relevant galanin neurons when comparing expression across neurons across the entire nervous system. Within the POA single cell dataset, *Magel2* displayed a two-fold higher expression in the galanin expressing parenting neuron type compared to the remaining POA neurons, making it a primary candidate for assessing parenting behaviour. We further showed a similar level of increased expression of *Magel2* in *Gal/Th* and *Gal/Calcr* cells of the POA via RNAscope when compared to other cells in the region. *Magel2*'s in-situ enrichment in *Gal/Calcr* cells was particularly dramatic and others [29] found *Gal/Calcr* cells to show the strongest *c-Fos* response following parenting.

As a key Prader-Willi syndrome gene, mice null for paternal *Magel2* have been extensively characterised, with phenotypes seen in metabolism [42], feeding [37], and several deficits in neonates including suckling [43] and USV production [36, 44]. Here we have shown that loss of *Magel2* results in deficits in parenting behaviour independent of pup USV production. USVs have been shown to affect maternal care at later developmental stages [36] but at the time window of this study, they appear to not have an impact on this retrieval paradigm. We saw parenting deficits in mothers, fathers and virgin female mice, and these mice have overlap in the circuitry necessary to produce parenting behaviour, with all relying on the same galanin POA hub [28]. However, whereas mothers are primed by the hormonal events of pregnancy in advance of experience [45, 46], fathers are dependent on post-mating cohabitation with pregnant females (minimum 2 weeks) to transition their virginal infanticidal behaviour to reliable parental care while their pups mature [47–49]. Virgin females on the other hand display 'spontaneous maternal behaviour' in which a certain proportion will display full maternal behaviour towards pups when first exposed [38, 39]. This proportion steadily increases upon subsequent exposures until 100% of these animals will display some parenting behaviour, although not to the same level of motivation and reliability as mothers. Finding deficits in one group but not others is a window into potential mechanisms. For instance, deficits only in mothers would suggest disruption to the priming effect of pregnancy hormones. However, deficits in all three groups, as seen here, suggests either a central mechanism, or multiple overlapping mechanisms, are affected in the *Magel2*-null mice. Due to the expression pattern of *Magel2* in the hypothalamus, we predicted one mechanism would be due to differences in the number or performance of POA galanin neurons.

We tested this idea and showed that *Magel2*-null females have a reduced *c-Fos* response in the POA following pup exposure confirming that *Magel2*'s action is impacting the POA's response to pup cues. We also saw that *Magel2*-null females have a clear reduction in the number of *Gal*-positive cells (and hence *Gal/Calcr* cells) and when the differing number of galanin

expressing cells was accounted for, there was no difference in the *c-Fos* activity of *Gal* cells between *Magel2*-null and WT. This suggests that *Magel2* is in part responsible for proper galanin expression in the POA and aligns with recent proteomic evidence in the hypothalamus of *Magel2*-null mice showing that galanin protein is down-regulated [50]. The exact mechanism by which *Magel2* would regulate galanin is still not clear, but *Magel2* is known to bind to a Wash Complex that regulates packaging and trafficking of neuropeptides including targets such as *Avp*, *Prl*, *Sst* and *Gal* [50] and removing *Magel2* from this system could result in insults to neuropeptide trafficking leading to a reduction in parenting-associated galanin neurons in the POA. Whether this loss of galanin is specific to the POA or a nervous system wide phenomenon was not investigated and whether other imprinted genes will mirror *Magel2*'s effect on POA galanin is an interesting avenue to explore in the future. Of note, a POA galanin mechanism does not prevent other mechanisms from acting and it is known that deletion of *Magel2* [43] and *Peg3* [13] results in a loss of oxytocin neurons in mice, neurons which feed directly into the POA and the loss of which could be responsible for deficits in certain facets of parenting behaviour.

Our systematic analysis of imprinted gene expression, and behavioural/molecular biology analysis of an exemplar candidate, taken together with our previous findings [30], strongly support the idea that parental care is indeed a physiological focus for genomic imprinting. Interestingly, loss of paternal *Magel2* in pups leads to reduced USV production which in turn impacts upon solicitation of parental care from wild-type mums [36], supporting the idea that genomic imprinting has evolved to coordinate the activity of parenting between mother and offspring [19, 20]. However, whether this evolutionary drive arises as a consequence of coadaptation between mother and offspring, or intragenomic conflict between parental genomes, remains an area of ongoing debate [51–53]. Nevertheless, our data demonstrate the importance of imprinted genes generally in influencing parental care and suggests that parental behaviours could be an evolutionary driver for genomic imprinting in the postnatal brain.

## Methods

### Ethics statement

Animal studies and breeding were approved by the Cardiff University's Ethical Committee and performed under a United Kingdom Home Office project license (PP1850831, Anthony R. Isles).

### Single-Cell RNA Seq Analysis of POA data (generated by [29])

Our full bioinformatic workflow has been described previously [30] but in brief, POA sequencing data were acquired through publicly available resources (GEO Accession–GSE113576) and the dataset was filtered and normalised according to the original published procedure. Cell identities were supplied using the outcome of cell clustering carried out by the original authors, so that each cell included in the analysis had a cell-type or identity. Positive differential expression between identity groups were carried out using one-sided Wilcoxon rank-sum tests (assuming the average expression of cells within the current identity group is 'greater' than the average of cells from all other groups). The test was performed independently for each gene and for each identity group vs. all other groups. The large number of $p$ values were corrected for multiple comparisons using a horizontal Benjamini-Hochberg correction, creating $q$ values. Fold-change (FC) values, percentage expression within the identity group and percentage expressed within the rest were also calculated. We considered genes to be significantly positively differentially expressed (significantly upregulated) in a group compared to background expression if it had a $q \leq 0.05$ and Log2FC $> 1$. The same custom list of imprinted

genes as the previous study were used (S3 Table). Enrichment was calculated using an Over-Representation Analysis (ORA) via a one-sided Fisher's Exact Test ('fisher.test' function in R core package 'stats v3.6.2'). The aim was to assess whether the number of imprinted genes considered to be upregulated as a proportion of the total number of imprinted genes in the dataset (passing the 20-cell filter) was statistically higher than would be expected by chance when compared to the total number of upregulated genes as a proportion of the overall number of genes in the dataset. To limit finding over-represented identity groups with only a few upregulated imprinted genes, an identity group was required to have $\geq$ 5% of the total number of imprinted genes upregulated for ORA to be conducted. Subsequent p-values for all eligible identity groups were corrected using a Bonferroni correction. This provided a measure of whether imprinted genes are expressed above expectation (as opposed to the expression pattern of any random gene selection) in particular identity groups. Mean fold change of expression for imprinted genes and for the rest of the genes within a subpopulation was also calculated.

Imprinted gene expression data for the Mouse Brain Atlas [31] and Whole Hypothalamus [32] were produced in our previous analysis [30] and all files can be found at our OSF repository (https://osf.io/jx7kr/) but are also provided as Supplemental Data. Imprinted gene expression data can be found as an 'Upregulated_IGs.csv' file for each analysis.

## Mice

All mice were housed under standard conditions throughout the study on a 12 h light–dark cycle with lights coming on at 08.00 h with a temperature range of 21˚C ± 2 with free access to water (tap water), and standard chow. Mice were either Wildtype (WT) on a C57BL/6J background acquired from Charles River Laboratories, or *Magel2*-null mice (paternal transmission of ablated allele) and their WT littermates on a C57BL/6J background. *Magel2*-null mice carried a constitutive deletion derived from a *Magel2*-FLOX-EM1.1 line generated by the Mary Lyon Centre (MLC) at MRC Harwell, Oxford UK. Genomic QC on this line confirmed correct deletion within the single exon and loss of expression in P4 brain was confirmed by qPCR (Fig F in S1 Text).

## Parenting behaviour assessment

**Subjects.** Fig 8A demonstrates the various experimental cohorts that underwent a parenting behaviour assessment. 25 male and female *Magel2*-null[(+/p)] mice were paired with WT mice generated in the same set of litters (but never mice from the same litter). A further 20 male and female WTs were also paired to be tested as a WT Control cohort. A separate cohort of virgin/naïve females, comprising 20 *Magel2-null*[(+/p)] and 20 WT virgin females, was also assessed. Mice were tested with the same 3 pups for their assessment. For mothers and fathers, this was three pups from their own litters. Hence mothers and fathers were only included in our assessment if they successfully produced a litter with at least 3 pups which survived until the end of the testing window (P5). For virgin females, litters were produced from separate WT x WT crosses (Charles River C57BL/6J) and each virgin female was assessed with a unique set of 3 pups.

To minimise the number of *Magel2-null*[(+/p)] pups used for testing (and hence able to manipulate parenting behaviour through altered behaviour), we selected the three heaviest pups (at P2) for every subject (which were distinguished on subsequent days by colouring the back of the pups with marker pen). Retrieval was performed outside of the time window in which disrupting *Magel2* expression has been shown to affect USV's [36, 44] and was performed under standard 21˚C conditions.

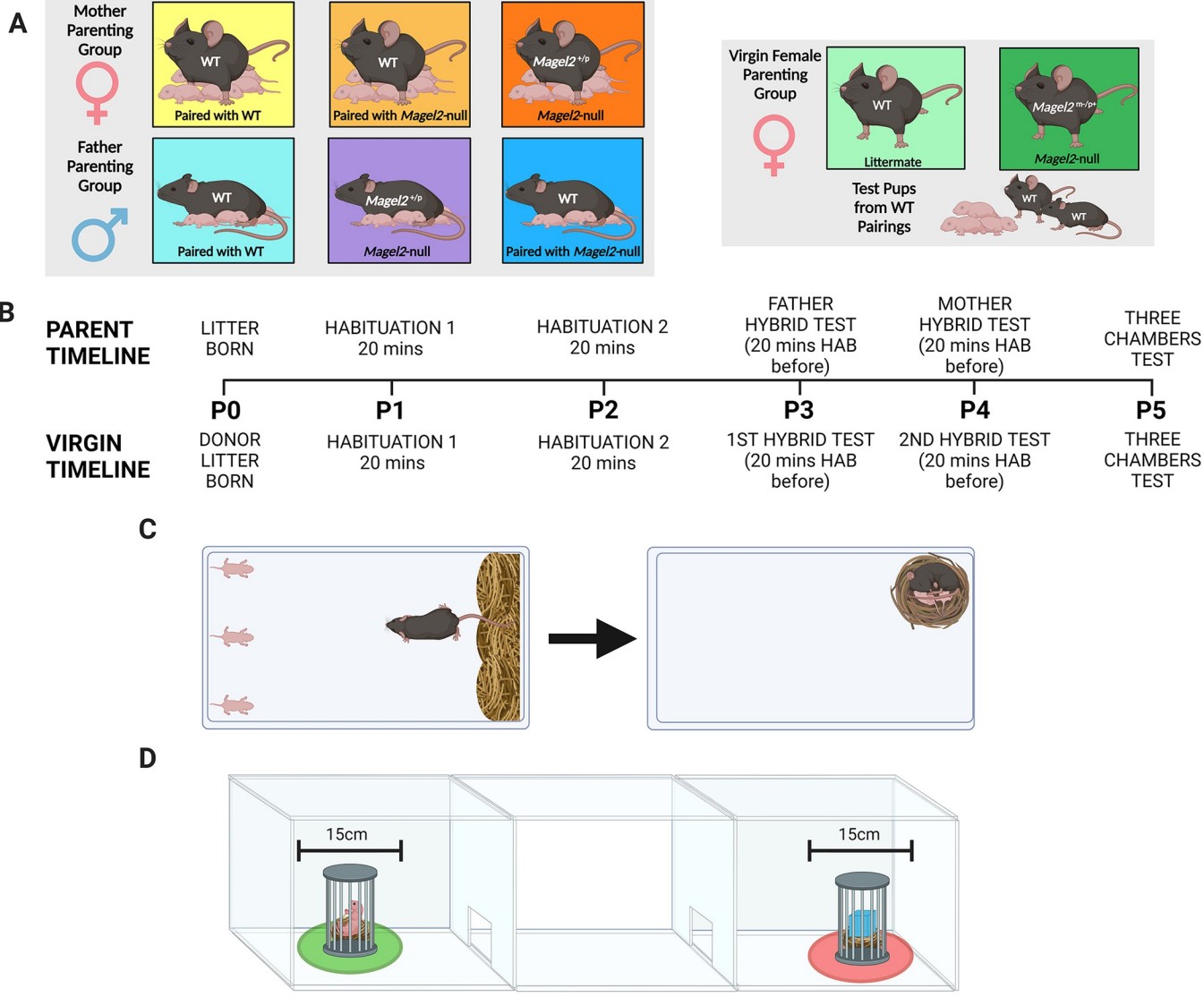

**Fig 8. Behavioural Paradigm and Set up. (A)** Behavioural Paradigm for parenting assessment in Mothers (Left), Fathers (Centre) and Virgin Females (Right). The maternal and paternal cohorts were paired with each other as indicated and tested with their own litters; Virgin females were tested with donor pups from separate WT pairings. **(B)** Timeline indicating the order in which tests and habituations were carried out based on the day the test litter was born. Above the timeline are events occurring for mothers, fathers and virgin females but alternative events specific to virgin females are indicated below. **(C)** Retrieval/Nest Building Combined Test. *Left*–set up pre-recording with 3 pups displaced to one short side of the cage and the home nest deconstructed against the opposite short side. *Right*–example of finished behavioural test with all three pups retrieved and visible nest re-constructed from the scattered material. **(D)** Three Chambers Test. Three pups used in retrieval are placed under a protective cage in one side chamber and a novel object is placed in an identical cage in the opposite side chamber. Time spent in a 15cm zone around the pup and novel object cage was measured for the pup-preference and pup-aversion scores. Created with Biorender.

## Tissue collection/genotyping

Following behavioural assessment, tail clips were taken from the all the pups used in the behavioural assessments. Genotyping was carried out by MRC Harwell to confirm the number of paternally inherited mutants were present in the testing pup population.

## Behavioural methods

All behavioural tests were carried out during the light phase of the light cycle (between 08:00–20:00) in a dimly lit room (< 30 lux). All mice were handled using the tunnel technique to avoid undue stress before and after the tests. Fig 8B shows the order in which the tests were carried out following the birth of a viable litter. All mice carried out a retrieval/nest building assessment followed by a three chambers assessment on a subsequent day. Following habituation, fathers performed the retrieval assessment when test pups were P3, mothers on P4 and both carried out the three chambers test on P5. Virgin females performed the retrieval assessment when test pups were P3 and P4 and also carried out the three chambers assessment on P5.

## Retrieval and nest building assessment

Animals were tested within their home cage which was cleaned to the same standard 2 days prior to testing (the day after the litter was born) by replacing soiled bedding and sawdust. For the test, the home cage had all enrichment removed and was placed, with metal cage lid removed, inside a plastic storage container. A HD webcam was attached to a table mounted tripod and set up, so it was positioned directly above the centre of the home cage. All mice received the same amount of nesting material to build their home nest from before their litter was born (one nest disc and one nestlet).

Directly prior to the test, the test animal (mother/father/virgin female) was removed from the home cage and placed in a holding cage. The other animal in the pair was then placed in a new clean cage with the enrichment from the home cage, a fresh nest disc and all the pups bar the three test pups. The three pups remaining in the home cage were positioned against one of the short ends of the cage (the end opposite the home nest–see Fig 8C) with two in the corners and one directly in between these two. The home nest was shredded completely and placed all the way along the opposite side of the cage from where the pups were placed (the side in which the home nest was previously located). Recording was started and test animals were returned to their home cage, placed directly onto the shredded nest. Animals had one-hour total time in which to complete the behaviour test. The goal of which was to retrieve the three scattered pups to the nest material and to re-construct the home nest using the scattered material.

## Mothers and fathers

Mice were paired together aged 9–12 weeks, females were weighed periodically to confirm pregnancies and litters were born when mice were aged 12–17 weeks. The day the litter was born was considered P0. On day P1 and P2, the home cage (with the mother, father and pups) was carried to the test room and placed in the testing apparatus with the camera suspended overhead for habituation. Both habituations lasted 20 minutes, on P1 the cage lid was left on and on P2 it was removed. Prior to the tests on P3(father)/P4(mother), the animals (mother, father, pups) underwent a 20-minute habituation period but this time the cage lid and all enrichment were removed from the cage. Following this the test was set up (as specified above) and began. The non-test animal and the remaining pups were removed from the test room before testing began.

## Virgin females

Virgin female mice were housed in same genotype pairs. When a litter was born (from the WT x WT matings), and 3 pups assigned to the virgin females, they were both habituated on day P1 and P2 (identically to the mothers and fathers). The virgin females were then tested twice, once on P3 and once on P4 (each time proceeded by the 20-minute habituation). The pair of

virgin females in a cage were tested one after another, the order of testing was reversed on the second day. Females who had been exposed to pups were kept isolated and not reintroduced to females that hadn't been exposed until their test was also carried out.

### Three Chambers Pup Preference assessment

On P5, all animals carried out a three chambers assessment with the same three pups as used in the retrieval test. The three chambers apparatus consisted of a white Perspex arena (40 x 30 x 30 cm, h x w x d) divided into three equal chambers connected in a row. Two Guillotine doors (5x5cm, operated by a pulley system) were used to connect each of the exterior chambers to the middle chamber.

For the pup preference assessment, mice were initially habituated to the middle chambers for 5 minutes with the guillotine doors closed. After 5 minutes, 3 pups (and fresh bedding) were placed at the outer edge of one of the exterior chambers with a protective cage placed over the top of the pups and weighted down. At the outer edge of the other exterior chamber, a novel object (a large Lego brick and an equivalent amount of fresh bedding) was placed and covered with an identical protective cage (See Fig 8D). The two Guillotine doors were then opened simultaneously. From this point, the mouse had 10 minutes in which to freely explore all chambers. Apparatus was wiped down thoroughly between trials and the chamber with pups in was alternated between trials.

### Behaviour metrics

The Retrieval and Nest Building combined test was scored at the millisecond level using Boris and provided several metrics. The timing for the tests began from the instance that the test mouse first sniffed any of the pups in the trial. From here, we recorded the time taken to retrieve each of the pups to the nest area, the time taken to construct a nest of sufficient quality and the time taken to complete the trial (have retrieved all three pups to a suitable quality nest). We scored the quality of the final nest built (on a 1–5 scale, [54]) by the time the trial had finished (or when the one-hour time limit had expired) (See Fig E in S1 Text for exemplar images of nest build quality scores). We also scored the amount of time that the animals spent performing pup-directed behaviour, defined as any of the following: sniffing pups, licking pups, grooming pups, carrying pups, nest building while pups are inside the nest, crouching/sitting in the nest while pups are inside. This was then scored as a proportion of the total time it took animals to retrieve all three pups and to finish the task. All metrics were scored blind of genotype by the primary scorer and 80/210 videos were also second scored by a second blind researcher. Interclass correlations coefficients on all metrics were greater than 0.75 with most scoring greater than 0.9 indicating a good/excellent level of agreement between the primary and secondary scorer (Table G in S1 Text).

The three chambers assessment was scored using Ethovision. We assessed the number of seconds that the test mice spent in the pup chamber compared to the object chamber, but more importantly, we recorded the time the animal spent within a 15cm diameter of the pup cage as well as the time spent within a 15cm diameter of the novel object cage. Motility analysis was also carried out using Ethovision. Velocity was calculated during the retrieval and three chambers' tests as well as the number of chamber crosses that each cohort performed during the three chambers test.

### Statistics and figures

For behavioural measures for Mothers and Fathers, all continuous variable analyses were performed using one-way ANOVAs and post-hoc pairwise t tests if the data met normality

assumptions while Virgin Females were analysed using two-way Mixed ANOVAs with Genotype and Exposure as variables. If normality assumptions were not met, then log transformations were used. If date were deemed non-normal/categorical, analysis was performed using the Kruskal Wallis Test followed by pairwise Dunn Tests or via the R package nparLD [55] which provides a rank-based alternative for analysing longitudinal data in factorial settings, Proportion variables were corrected using an arc sine correction.

### RNAscope quantification of Magel2 and c-Fos in Gal/Calcr/Th cells in the POA

Fig 9 displays a simplified workflow for the RNAscope from tissue harvest to data analysis.

### Pup-exposure behavioural paradigm

For the *c-Fos* quantification, mice were either *Magel2*-null (*Magel2*$^{(+/p)}$, N = 16) or Wildtype littermates (WT, N = 16) on a C57BL/6J background. For each sex, four mice from each genotype (WT and *Magel2*-null) were permanently cohoused with a WT non-sibling mate to produce litters. All mice produced litters with 3+ pups. When pups were P1 –P2, test mice were habituated to the test room by placing the cage in the test room for 30 minutes with mice and pups in the cage. On test day (P3 for Males and P4 for Females), pups and mates were removed from the home cage and housed in a new clean cage. Pup-exposed animals were then left for a one-hour isolation period in the home cage, in the test room they had been habituated, to standardize the *c-Fos* activity in the POA. During this hour, the mice were exposed to no stimuli in the form of pups, other mice or the experimenter. After the hour, 3 pups were transported to the test room and returned to the home cage with the test animal, placed on the opposite side of the cage from the undisturbed home nest. The 30 minute exposure began when the test animal first investigated the pups. 30 minutes later, pups were removed, and maternal/paternal pup-exposed mice were transported for perfusion and tissue harvest. Non-pup-exposed WTs (N = 4 per sex) and *Magel2*-nulls (N = 4 per sex) were habituated in the same manner, isolated for one hour in the test room and transported directly from the home cage for tissue harvest. For *Magel2* RNA quantification experiment, WT animals (3M, 3F) were used straight from the home cage without an isolation period.

### FFPE Tissue preparation

Once ready for tissue harvest, all animals were transcradially perfused with 10% Neutral Buffered Formalin (NBF) before whole brains were taken. A 3mm section of tissue was taken using a brain slicing matrix with 1mm slice channels (Zivic Instruments). This 3mm section was taken with the POA situated in the centre of the block. 3mm sections then underwent standard pre-paraffin embedding procedures and were sectioned at a thickness of 10μm with every 8$^{th}$ section used for H&E staining and every 9$^{th}$ and 10$^{th}$ section mounted on one slide for RNA Scope. Only sections containing the POA were subsequently analysed. For *Magel2*-quantification, two animals (1M,1F) were assessed for *Gal/Th* neurons and four animals (2M,2F) were assessed for *Gal/Calcr* neurons. For *c-Fos* quantification, only sections containing high numbers of *Gal/Calcr* cells (identified from the previous experiment) were subsequently analysed (4–5 sections per animal).

### RNAscope protocol

Three-plex RNA Scope was performed using RNAscope Multiplex Fluorescent Reagent Kit v2 (ACD Bio-techne) on FFPE brain sections. The manufacturer's protocol was followed exactly

1. **Tissue Harvest**
- 3mm block with POA taken
- Preprocessed and Paraffin Embedded

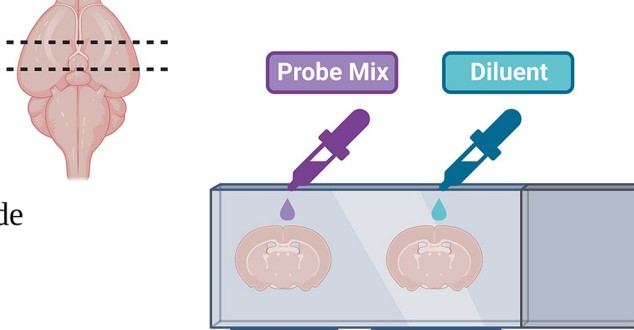

2. **Sectioning**
- Sectioned at 10um
- Every 9th and 10th section taken on one slide

3. **Three-plex RNAscope**
- Following standard procedure
- One section received RNA target probes (Probe Tissue) while other received diluent (No Probe Control).

4. **Image Acquisition**
- Both sections imaged together
- Same light intensity/duration per gene/channel for every slide

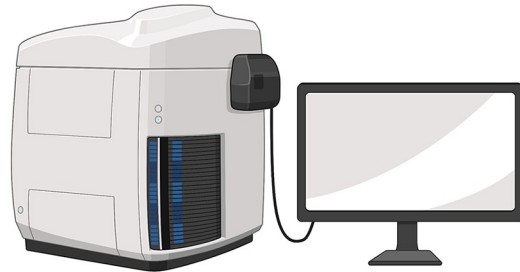

5. **Image PreProcessing**
- Background Subtraction - Rolling ball (50-pixel)
- Smoothing - Gauss (DAPI = 2px, Channel = 1px)

6. **Threshold quantification for each channel**
- No Probe Control Tissue (diluent)
- ROI defined manually
- DAPI defined nuclei + 25μm cytoplasm
- Threshold Calculated per channel
  - Avg. Max intensity within a cell + 3SD

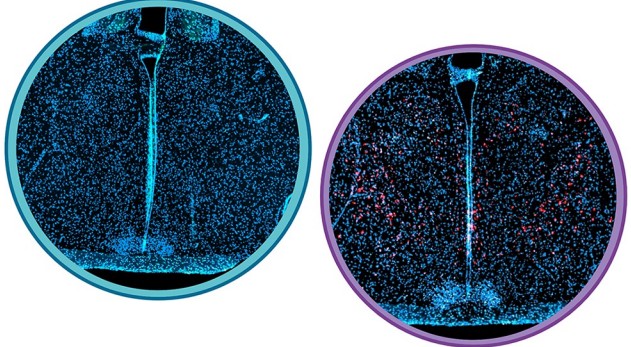

7. **Signal quantification for each channel**
- Probe Tissue - ROI defined manually
- DAPI defined nuclei + 25μm cytoplasm
- All signal above threshold within cytoplasm
- Clusters resolved into molecule counts

8. **Analysis**
- Target cells vs. Rest
- Quantitive - No. Target RNA molecules
- Semi-quantitive - H-Score of Target RNA molecules

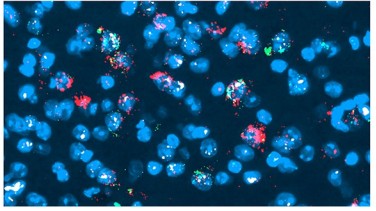

**Fig 9. Summary of RNAscope image analysis workflow.** A 3mm block of mouse brain was harvested, fixed and paraffin embedded. Every 9th and 10th section through the block were taken on the same slide. Both tissue sections underwent the full RNAscope protocol however during the addition of the probe mix, one section was the experimental tissues and received a probe mix (either *Gal/Th /Magel2*, *Gal/Calcr/Magel2* or *Gal/Calcr/Fos* probe mix) and the other received diluent during this step, to act as our no-probe control. Images were acquired on a Zen AxioScan Z1 at 20x magnification and, fluorescent light intensity and duration were kept the same between slides of the same probe mix. Images were pre-processed. The POA of the no-probe control tissue was defined, nuclei resolved, cytoplasm defined and then the maximum intensity of a pixel for each channel within each cell was recorded. This value indicated the minimum threshold needed for this cell to be classed as positive for that gene. Each gene then had a threshold value derived using the average maximal intensity plus three standard deviations. This threshold value was applied to the probe tissue to identify signal. Clusters were resolved and gene probe expression was analysed quantitively and semi-quantitively as advised by the manufacturer. Created with Biorender.

following the 'standard' pre-treatment guidance. Briefly, slides underwent deparaffinization and H202 incubation before being added to boiling RNAscope Target Retrieval for 15 mins, followed by a 30 mins incubation at 40˚C in the HybEZ Oven with Protease Plus. Probes were incubated for 2 hours at 40˚C in the HybEZ Oven. One section on the slide received the probe mix while the other section on the slide (and hence containing the adjacent cells) received an equal amount of probe dilutant to act as a no-probe control. Amplification steps then followed before each channel's signal was developed. Tissue either received a *Gal/Th/Magel2* probe mix, a *Calcr/Gal/Magel2* probe mix or a *Calcr/Gal/Fos*. Hence HRP-C1 corresponded to either the *Gal* Probe (ACD 400961, Mm-*Gal*) paired with the fluorophore TSA Vivid 520 or the *Calcr* Probe (ACD 494071,Mm-*Calcr*) paired with TSA Vivid 570. HRP-C2 corresponded to the *Th* Probe (ACD 317621-C2, Mm-*Th*-C2) paired with TSA Vivid 570 or the *Gal* Probe (ACD 400961-C2, Mm-*Gal*-C2) paired with TSA Vivid 520. HRP-C3 corresponded to the *Magel2* Probe (ACD 502971-C3, Mm-*Magel2*-C3) or the *Fos* Probe (ACD 316921-C3, Mm-*Fos*-C3) and TSA Vivid 650 was assigned to this channel. All fluorophores were applied at a concentration of 1:1500. Slides were counterstained with DAPI (30 seconds) and mounted with Prolong Gold Antifade mounting medium.

## Image acquisition and analysis

Whole brain slides were imaged within one week of mounting using Zeiss AxioScan Z1 at 20x magnification with the same light intensity/duration settings used for each scan. Images were analysed with Zen Blue 3.6 (See Table H in S1 Text for acquisition, processing and analysis settings). Briefly, images were pre-processed using a 50-pixel radius rolling ball background correction followed by Gauss smoothing (2 pixel for DAPI, 1 pixel for the other channels). For both sections on the slides, the POA was manually defined as the ROI. The no-probe-control section (the adjacent section on the slide with probe diluent instead of probes) was analysed first. Using Zen Blue's Image Analysis, nuclei within the ROI were localized using DAPI signal intensity and a 25um border was placed around each nucleus as an estimated cytoplasm. For every cell (nucleus plus cytoplasm) in the no-probe control, the intensity of pixels within a cell were quantified and the maximum intensity value and average intensity values for each of the channels were calculated. The maximum intensity value for each cell in the no-probe control can be considered the value by which that cell would be considered positive if that value was set as the threshold. We hence used this value to calculate a threshold value for each channel, by taking the average maximum intensity value for cells in the no-probe control plus three standard deviations.

Next, for the probe-sections, nuclei were again localized within the ROI using DAPI signal intensity (with the same settings) and a 25um cytoplasm. Fluorescent pixels within this 25um border which exceeded the threshold value for that channel found in the control were counted for each channel. Signals from all genes quantified here tended to display clusters as well as individual signals. Clusters were resolved into molecule counts using the guidance from ACD Bio-techne (SOP 45–006). Specifically, the average integral intensity of individual signals (minus average background) was calculated, and clusters were resolved by divided the integral intensity of the cluster (minus average background) by this average. The finished data took the form of individual molecule counts for each of the channels for each DAPI identified cell.

## Statistics and figures

*Magel2* RNAscope image data were analysed in two ways. Firstly, our quantitative analysis used molecule counts for *Magel2* were compared between *Gal/Th* & *Gal/Calcr* vs. the rest of the preoptic area cells using either Wilcoxon Ranked Sums Tests with Bonferroni correction

or Kruskal Wallis Tests with post hoc Bonferroni corrected Dunn tests for multiple comparisons. Secondly, ACD Bio-techne also suggested a semi-quantitative metric by deriving a H-Score for the groups compared. The percentage of cells: with 0 *Magel2* molecules was multiplied by zero, with 1–3 molecules was multiplied by one, with 4–9 molecules was multiplied by two, with 10–15 molecules was multiplied by three and with 16+ molecules was multiplied by four. The resulting H-Scores are reported for all comparisons in Table D in S1 Text.

For c-*Fos* RNAscope image data, we compared the proportions of positive cells for a particular gene (*Gal/Calcr*– 2+ molecules, *Fos*– 5+ molecules) between areas of the brain and cell types. Variability in sections/POA position was accounted for by normalizing *Fos/Gal/Calcr* positive cell counts per animals to counts per 1000 POA cells. WT and *Magel2*-nulls, pup-exposed and non-pup-exposed, males and females, were compared using three-way ANOVA. Additionally, the average number of gene molecules was also compared between *Magel2*-null mice and WTs and analysed with three-way ANOVA also.

For all experiments, graphical representations and statistical analyses were conducted using R 3.6.2 [56] in RStudio [57].

## Supporting information

**S1 Text. Supplemental Methods: i. Medial Preotic Area dataset. ii. Mouse Brain Atlas dataset. iii.** Whole Hypotahlamus dataset. Figs 1–6 in S1 Text; Tables A-H in s S1 Text. (DOCX)

**S1 Table. Cell marker genes used in the three different scRNA-seq datasets.** (XLSX)

**S2 Table. ANOVA output for *Gal*, *Calcr* and *Fos* RNAscope data.** (XLSX)

**S3 Table. Imprinted gene list used in enrichment analysis.** (XLSX)

## Acknowledgments

We would like to thank the research groups that carried out the single-cell RNA sequencing that were utilised in this study. We express further gratitude to the Cardiff Bioimaging Hub for embedding and sectioning the brain samples used for RNA Scope and to Alice Chibnall for RNAscope training.

## Author Contributions

**Conceptualization:** Matthew J. Higgs, Rosalind M. John, Anthony R. Isles.

**Data curation:** Matthew J. Higgs, Anna E. Webberley.

**Funding acquisition:** Anthony R. Isles.

**Investigation:** Matthew J. Higgs, Anna E. Webberley, Alasdair J. Allan, Moaz Talat.

**Methodology:** Matthew J. Higgs, Anthony R. Isles.

**Supervision:** Anthony R. Isles.

**Writing – original draft:** Matthew J. Higgs, Anthony R. Isles.

**Writing – review & editing:** Matthew J. Higgs, Rosalind M. John, Anthony R. Isles.

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
