## [Decision Letter · Decision Letter 0]

17 May 2023

Dear Dr Isles,

Thank you very much for submitting your Research Article entitled 'The parenting hub of the hypothalamus is a focus of imprinted gene action' to PLOS Genetics.

The manuscript was fully evaluated at the editorial level and by independent peer reviewers. The reviewers appreciated the attention to an important problem, but raised some concerns about the current manuscript, which should be relatively straight-forward to address. Based on the reviews, we will not be able to accept this version of the manuscript, but we are looking forward to review a revised version. We cannot, of course, promise publication at that time.

If you decide to revise the manuscript for further consideration at PLOS Genetics, please aim to resubmit within the next 60 days, unless it will take extra time to address the concerns of the reviewers, in which case we would appreciate an expected resubmission date by email to plosgenetics@plos.org.

We are sorry that we cannot be more positive about your manuscript at this stage. Please do not hesitate to contact us if you have any concerns or questions.

Yours sincerely,

Marisa S Bartolomei

Academic Editor

PLOS Genetics

Wendy Bickmore

Section Editor

PLOS Genetics

Reviewer's Responses to Questions

**Comments to the Authors:**

Reviewer #1: This paper reveals the impact of imprinted genes in parenting based on the expression level of these genes in galanin-expressing POA neurons, known to be involved in parenting. The study then focused on the imprinted gene Magel2, confirming the high level of expression in galanin neurons and investigating the role of Magel2 in parental behaviour. Magel2 KO mothers, fathers and virgin females showed alterations in parental behaviour. Magel2 KO females show a reduction in the number of galanin neurons and a reduction in the number of these cFos positive cells after pup-exposure , suggesting a decrease in the neuronal activity of galanin neurons.

This is an interesting article, very well written. The genetic control of parental behaviour, both maternal and paternal, is a very interesting and yet little studied topic. This article reveals a particular role of imprinted genes in this behaviour and thus proposes a new functional role of this gene family in the parent-child relationship. The research is well conducted using single cell RNA-seq analysis, RNAscope studies and behavioural studies to assess parental behaviour. The results are clear and the data analyses are robust.

Major comments:

-The Magel2 null mouse line is identified as "Magel2-FLOX-EM1.1". It is necessary to describe this mouse line if it is a novel line or to give a reference (publication) where the mouse line is described.

-The paragraph concerning cFos activity (line 348 to 382) is written taking into account, on the one hand, the genotype and, on the other hand, the sex. As it stands, it is difficult to read the comments and look at the figures. This paragraph should be written taking into account sex and genotype by comparing mutant males with the control and mutant females with the control (as in the figures), it will be easier to understand.

Minor comments :

-Figure S3b (B): There is no difference in WT exposed-pups males compared to WT controls but in Magel2-null there is a difference. Why is this?

-Additional data format: citations in the text of table S2A and B disrupt the text.

-It would have been interesting to have, as in females, the data on cFos expression in Galanine neurons in Magel2KO fathers and virgin females compared to WT fathers.

Reviewer #2: Higgs and colleagues utilized a systems biology approach on specific neuronal populations responsible for controlling parental care to investigate whether imprinted genes converge to regulate parenting behaviour. Using single-cell RNA sequencing datasets, they found an enrichment of imprinted gene expression in galanin-expressing neurons of the preoptic area, which is a brain hub for regulating parenting behaviour. Then, they focused on Magel2, an imprinted gene not previously associated with parenting behaviour. They confirmed its expression in preoptic area galanin-expressing neurons and then examined the parenting behaviour of Magel2+/- null mice. The results showed deficits in pup retrieval, nest building, and pup-directed motivation in Magel2-null mothers, fathers, and virgin females, revealing a central role for this gene in parenting. Additionally, they found a significant reduction in POA galanin-expressing cells in Magel2-null mothers and fathers, which contributed to a reduced c-Fos response in the POA upon exposure to pups.

Overall, this study identified a novel imprinted gene that affects parenting behaviour and demonstrated the usefulness of using single-cell RNA sequencing data to predict gene function from expression. It also revealed the important role of genomic imprinting in mediating parental behaviour.

The study presents a well-defined narrative and experimental strategy, which involves using a publicly available dataset to conduct a thorough analysis and identify the appropriate cell population target in mPOA. I found it noteworthy that the fathers were included in the parental evaluation, which adds a unique aspect to the study.

I have compiled a set of comments below, which I hope will provide the authors with helpful insights to enhance their manuscript.

Comments

The experiment with virgin females indicates that they may have improved their ability to perform the task during the second exposure, suggesting that the pup retrieval test may not necessarily reveal a deficiency in parenting behaviour. Instead, the mutant mice (mothers, fathers, and virgin females) may simply be slower to respond to unfamiliar situations created by the experimental conditions. Are there any differences in "normal" parenting observed in the home cage? Additionally, considering certain neural disorders like autism that do not necessarily result in poor parenting but can affect an individual's ability to cope with new and unfamiliar situations, do the authors believe that a naturalistic approach to assessing parental behaviour would produce different results?

In a study by Bosque Ortiz and colleagues (https://doi.org/10.1111/gbb.12776), it was reported that loss of Magel2 impairs separation-induced vocalization of the offspring and maternal behaviour towards these pups, which appears to be in contrast with the findings by Higgs. Can the authors comment on this? One possible explanation is that the time window considered by Higgs (P3-P5) precedes the one examined in the Bosque Ortiz paper.

The nest building and pup retrieval test is known to induce stress in mice. Given this, I am curious about how Magel2-mutant mice cope with stress and whether this plays a role in their performance. While the test itself is inherently stressful and difficult to limit in a traditional setting, I am wondering if the authors took any measures to reduce the stress effects.

The authors stated that the pup retrieval-nest building test was considered finished when all the pups were retrieved, and the nest was reconstructed to achieve a level 3 quality. If possible, I suggest they include a scale with relevant images of the different scores describing how the nest quality assessment is conducted, possibly in the supplementary section. Although Figure 8 provides some information, the addition of a more detailed scale with pictures would be beneficial.

Additionally, since the style of assembling the bedding material varies among mice, is it possible that the mothers in the Higgs study have differences despite all having the same background?

This paper by Ho-Shing and Dulac (2019, https://doi.org/10.1016/j.cobeha.2018.08.008) reports that loss of the paternally expressed Peg3 or Magel2 results in a reduction in hypothalamic oxytocin neurons. I am wondering if Magel2's effects on oxytocin releasing neurons might be responsible for some behavioural outcomes observed in this paper that seem to result from Magel2 defects in Gal+ neurons. For example, if there are fewer oxytocin releasing neurons, lactation might be impaired. In addition to keeping pups warm, crouching is also necessary in the lactation process. Observing a lower amount of crouching behaviour may result from Magel2 defects outside the Gal+ neurons identified in this study. What is the opinion of the authors?

As the study revealed an improvement in stranger performance during the second day of pup retrieval, I am wondering whether the same effect was observed in both mothers and fathers.

Minor comments

I find the mix of terms used to refer to the experimental groups, such as "Magel2+/−", "Magel2m+/p−", "Magel2 knockout", and "Magel2 null", to be confusing. I recommend that the authors use a consistent terminology and simplify how they refer to the groups.

Can the authors provide any information on the expression of Magel2 in the POA of Magel2m+/p− mice? Specifically, is there any expression from the maternal allele?

It may be useful to include a Venn diagram in Figure 1 to illustrate the overlap between genes such as Asb4, Calcr, Magel2, Ndn, Nap1l5, Peg3, Peg10, etc.

Caption Figure 1:

A) “a > 150%”: reads as a is greater than 150%

B) “elevated expression”: same as in A? expression level > 150%?

Figure 2A–B, 7A: missing scales

Figure 2C–D:

Can you explain the meaning of the error bars in this graph? Perhaps, a better visualization would be a violin or box plot with a swarm plot overlay to show the distribution of values, which would be consistent with the presentation in Supplemental Figure S1.

Figure 3, 4:

I suggest considering adding a panel that provides a description of the task.

Figure 3A, 4A:

Was the test conducted with only three pups? The figure seems to show more.

Figure 3C–E, G, 4C–E, G, 6C–E, G:

It could be misleading to include the mothers and fathers who were unable to complete tasks C, D, or E or build the nest in panel G. Also in Figure 5C–D

Figure 7:

It would be helpful to have a clearer explanation of the term 'pup exposure' in the experimental scheme, and to provide more details about the different periods of isolation (1 hour) and pup exposure (30 minutes) mentioned in lines 608-622.

The legend needs to be adjusted such that the colour of the points matches that in the figure, where Control points are grey. However, it is a bit confusing that the female and male points are represented by light grey and grey, respectively, which can be easily mistaken for the Control points. Therefore, it may be better to replace the legend with "Fos-positive cells per 1000 Gal/Calcr-positive POA cells" because much of the differences observed are due to the varying numbers of Gal/Calcr-positive cells (as shown in Figure 7D). For example, the difference between Magel2-null and WT females is solely due to the differences in the number of Gal/Calcr-positive cells, and the same applies to the difference between Magel2-null and WT males.

D–F) It is unusual that the bars in this figure are not divided as they were in Figures 7B-C, even though it is understood that no significant differences are expected.

C–F) In previous figures, the WT bars were always presented before the Magel2-null bars. It might be better to follow the same pattern here and in corresponding supplemental figures.

Caption Figure 7:

“5+ molecules”: Does it mean at least 5 molecules?

Text

Please maintain consistency in using the terminology "pup-directed behaviour" throughout the paper.

• L108–109: In contradiction Figure 1C seems to show that there are 10 upregulated imprinted genes in GABA10.

• L109: 2B –> S2B

• L118–119: language

• Lines 185, 188, 193 and others: “F2, 62”: should probably be H(2) or F(2, 62)?

• Line 329: PDB (also used in Figures 3I–J and 4I–J) not defined in text (possibly in line 199)

• Line 276: check if W = 117 correct like in Line 274 but with different p-value, also check number of digits in rounding

• Line 357–364: Unlike stated here, actually there is no effect of the Genotype on c-Fos expression (neither in Gal/Calcr-positive nor in Gal-positive nor in Calcr-positive cells), but rather on the number of cells expressing Gal and only subsequently c-Fos expression is influenced. The effect of the Genotype is only on the number of Gal/Calcr-positive, Gal-positive and Calcr-positive cells, but not on the c-Fos expression. The loss of c-Fos-positive cells is only for cells which are Gal-negative and Calcr-negative and only for females. Later the conclusion in Lines 377–378 states it correctly.

• Line 393–394: “Magel2-null females had reduced Gal expression in the POA and a reduced c-Fos response in the POA upon exposure to pups”:

◦ structure of sentence may be misunderstood: “reduced Gal expression upon exposure to pups”

◦ It seems that the sentence suggests that exposure to pups resulted in a decrease in c-Fos response, when in fact, the opposite is true. Perhaps the intention was to say that the increase in c-Fos expression is not as significant in Magel2-null females as it is in WT females. It would be helpful to clarify this in relation to Figure 7B (left). The figure shows an increase in c-Fos expression for both WT and Magel2-null females, but it would be beneficial to know the p-value for this difference.

Reviewer #3: The parenting hub of the hypothalamus is a focus of imprinted gene action

Here Higgs et al present an elegant study investigating the genomic imprinted gene expression in the hypothalamus and the role imprinting plays in parental care behaviours. Deletions of the imprinted genes Mest, Peg3 and Dio3 have previously been shown to influence parenting behaviours when deleted. Here the authors reassessed previously generated single cell data and find that imprinted gene expression is enriched in Gal-expressing neurons from the preoptic area of the hypothalamus, a neuronal sub-type known to regulate parental behaviour. They then assess parenting behavoiur in mice lacking Magel2, a gene they identified in their analysis that has not previously been shown to influence parental care giving. They find that Magel2 deficient mothers, fathers and virgin mice display perturbed parenting behaviours. They also find that Magel2 null females have reduced Gal expression in the POA and a reduced c-Fos response in the POA upon exposure to pups. This is a well-conceived and presented study, my comments are all minor.

Comments

Did the authors assess the binding specificity of the Magel2 probe in the null brains? These data would provide a control that they are interrogating the correct molecule in the RNAscope experiments and should be including in supplementary information if available

Line 107 - this sentence does not make sense.

Of interest were, GABA13 (Gal, Slc18a2 and Th) which showed enrichment for imprinted genes and GABA10 (Gal, Calcr and Brs3) (Supplemental Table 2B) but was not enriched for imprinted genes.

Figure 1 - It should be made clear what the genes listed in parentheses after each cell subtype are - I assume they are the defining neuronal markers for that cell type.

Line 184 - The time taken for to complete the task... The “for” should be removed.

For Supplementary Table 4B - Chapter 4 is mentioned in the legend. This should be removed.

**Have all data underlying the figures and results presented in the manuscript been provided?**

Reviewer #1: Yes

Reviewer #2: Yes

Reviewer #3: Yes

PLOS authors have the option to publish the peer review history of their article (what does this mean?). If published, this will include your full peer review and any attached files.

Reviewer #1: No

Reviewer #2: No

Reviewer #3: No

---

## [Decision Letter · Decision Letter 1]

7 Sep 2023

Dear Dr Isles,

We are pleased to inform you that your manuscript entitled "The parenting hub of the hypothalamus is a focus of imprinted gene action" has been editorially accepted for publication in PLOS Genetics. Congratulations!

Yours sincerely,

Marisa S Bartolomei

Academic Editor

PLOS Genetics

Wendy Bickmore

Section Editor

PLOS Genetics

Comments from the reviewers (if applicable):

Reviewer's Responses to Questions

**Comments to the Authors:**

Reviewer #1: The authors have responded to all the questions/changes suggested in my first review. Considering all the changes made in this article, it is now an excellent article.

Reviewer #2: none

Reviewer #3: The authors have addressed all of mine and the other reviewers previous comments. I have no further comments to make.

**Have all data underlying the figures and results presented in the manuscript been provided?**

Reviewer #1: Yes

Reviewer #2: Yes

Reviewer #3: Yes

PLOS authors have the option to publish the peer review history of their article (what does this mean?). If published, this will include your full peer review and any attached files.

Reviewer #1: **Yes: **Françoise Muscatelli

Reviewer #2: No

Reviewer #3: No

**Data Deposition**

http://datadryad.org/submit?journalID=pgenetics&manu=PGENETICS-D-23-00392R1

**Press Queries**

---

## [Editor Report · Acceptance letter]

22 Sep 2023

PGENETICS-D-23-00392R1 

The parenting hub of the hypothalamus is a focus of imprinted gene action 

Dear Dr Isles, 

We are pleased to inform you that your manuscript entitled "The parenting hub of the hypothalamus is a focus of imprinted gene action" has been formally accepted for publication in PLOS Genetics! Your manuscript is now with our production department and you will be notified of the publication date in due course.

With kind regards,

Judit Kozma

PLOS Genetics

On behalf of:
